# Extrusion of BMP2+ surface colonocytes promotes stromal remodeling and tissue regeneration

Julian Heuberger [1,2,5,6], Lichao Liu [1,6], Hilmar Berger [1], Joop van den Heuvel[3], Manqiang Lin [1,2], Stefanie Müllerke[1,2], Safak Bayram[1,2], Giulia Beccaceci [1,2], Hugo de Jonge [4], Ermanno Gherardi[4] & Michael Sigal [1,2] ✉

The colon epithelium frequently incurs damage through toxic influences. Repair is rapid, mediated by cellular plasticity and acquisition of the highly proliferative regenerative state. However, the mechanisms that promote the regenerative state are not well understood. Here, we reveal that upon injury and subsequent inflammatory response, IFN-γ drives widespread epithelial remodeling. IFN-γ promotes rapid apoptotic extrusion of fully differentiated surface colonocytes, while simultaneously causing differentiation of crypt-base stem and progenitor cells towards a colonocyte-like lineage. However, unlike homeostatic colonocytes, these IFN-γ-induced colonocytes neither respond to nor produce BMP-2 but retain regenerative capacity. The reduction of BMP-2-producing epithelial surface cells causes a remodeling of the surrounding mesenchymal niche, inducing high expression of HGF, which promotes proliferation of the IFN-γ-induced colonocytes. This mechanism of lineage replacement and subsequent remodeling of the mesenchymal niche enables tissue-wide adaptation to injury and efficient repair.

The colonic epithelium is exposed to a potentially toxic and bacteria-rich luminal environment. Several barrier mechanisms of epithelial "self-defense", such as mucus and antimicrobial protein production, ensure a selective permeability, shielding the epithelium from microbes while allowing nutrient and water uptake. Injuries to the epithelium can result in the loss of this protective shield, which can further exacerbate the injury and enhance the risk for systemic infections and intoxications. Therefore, injuries to the epithelium require rapid and efficient repair, which is facilitated by its high grade of plasticity. At homeostasis, Wnt-signaling-dependent Lgr5-expressing stem cells at the crypt-base steadily generate progenitor cells that further differentiate into absorptive or secretory cells[1]. BMP signaling has been shown to participate in this differentiation process[2–5]. In the

stomach, we demonstrated that surface epithelial cells establish a BMP-signaling feed-forward loop with the surrounding stromal cells, further reinforcing the differentiation process[6]. This intricate cellular hierarchy is lost after epithelial injury: Lgr5-expressing stem cells are frequently depleted, and regeneration is driven by various types of progenitors and even differentiated cells[7–12]. Regeneration, therefore, does not rely on Lgr5-expressing stem cells but on reprogrammed regenerative cells, which share features with the highly proliferative fetal epithelium and are characterized by high activity of YAP[13–15]. Yet, the mechanisms that drive the reorganization of epithelium upon injury and promote the acquisition of the regenerative state are not well understood.

We previously found that interferon gamma (IFN-γ) can interfere with the regulatory network that maintains homeostasis of the gastric

[1]Department of Hepatology and Gastroenterology, Charité—Universitätsmedizin Berlin, Berlin, Germany. [2]Berlin Institute for Medical Systems Biology (BIMSB), Max Delbrück Center for Molecular Medicine, Berlin, Germany. [3]Helmholtz-Zentrum für Infektionsforschung GmbH, Braunschweig, Germany. [4]Immunology and General Pathology Unit, Department of Molecular Medicine, Università di Pavia, Pavia, Italy. [5]Present address: Department Experimental Toxicology and ZEBET, German Federal Institute for Risk Assessment, Berlin, Germany. [6]These authors contributed equally: Julian Heuberger, Lichao Liu. ✉e-mail: michael.sigal@charite.de

epithelium[6]. IFN-γ is a major cytokine released by subsets of T cells that are recruited to injured epithelium[16]. Different effects of IFN-γ on intestinal epithelial cells have been suggested: In the small intestine, IFN-γ can disrupt the stem cell niche and cause a loss of stem cells and adjacent Paneth cells[17–19] and may enhance acute colitis[20]; in organoids, it induces colonocyte differentiation[21]. By contrast, in mice that lack IFN-γ signaling, the epithelium fails to acquire the regenerative fetal-like state upon parasitic infection[22]. We thus aimed to explore the role of IFN-γ during epithelial damage and regeneration in the colon.

We find that IFN-γ acts as a central driver of crypt reorganization upon colonic injury: IFN-γ induces rapid apoptosis and extrusion of differentiated surface colonocytes that express high levels of *Bmp2*. Simultaneously, the stem and progenitor cell pool differentiate towards distinctive colonocytes that do not express *Bmp2*. The reduction of epithelial-derived BMP-2 results in reduced activation of BMP signaling in the adjacent stromal cell compartment, causing a reprogramming of this compartment and high expression of regenerative factors such as hepatocyte growth factor (HGF). HGF, in turn, promotes the proliferation of the IFN-γ-induced colonocytes, enabling their entry into the regenerative state. Our findings represent a previously unappreciated principle of tissue adaptation to injury.

## Results

### Injury reduces WNT activity via IFN-γ
To study the role of IFN-γ in the context of colonic epithelial injury, we induced epithelial damage in vivo by exposing *IFN-γR receptor knockout (IFN-γR KO)* mice and wild-type (WT) littermates to dextran sodium sulfate (DSS) for 5 days, as described previously[12,23]. DSS disrupts the mucus layer and damages the colonic epithelium, inducing a strong inflammatory response with the influx of IFN-γ-producing T cells[24]. In situ hybridization (ISH) for the Wnt-target genes and stem cell markers *Axin2* and *Lgr5* revealed widespread crypt cell responses along the length of the colon, even beyond the strongly affected ulcerated areas, including areas with an intact crypt structure (Fig. 1a, b). In the homeostatic epithelium, *Axin2* and *Lgr5* were expressed predominantly in the base of the crypt, and the pattern of expression did not differ between WT and *IFN-γR KO* mice. As described previously[12], in the WT mice, DSS caused a strong reduction of *Lgr5* and *Axin2*. In *IFN-γR KO* mice, by contrast, expression of *Lgr5* and *Axin2* remained unchanged (Fig. 1a, b, and quantification in c, d). Thus, we conclude that IFN-γ-signaling suppresses the Wnt and stem cell marker genes *Lgr5* and *Axin2* in the colonic crypt upon injury.

### KRT20+ cells are enriched in colitis via IFN-γ
To assess whether IFN-γ-signaling also affects the differentiated cell compartment, we analyzed the areas with an intact crypt structure for expression of the colonocyte marker KRT20 using immunofluorescence. At homeostasis, the abundance of KRT20 expression was comparable between WT and *IFN-γR KO* mice. Most of the KRT20 signal was restricted to surface colonocytes. However, during acute colitis, the abundance of KRT20-expressing cells was increased, with WT mice showing a much greater increase than *IFN-γR KO* mice (Fig. 1e–f). In contrast, alcian blue staining of differentiated goblet cells revealed a strong reduction of the secretory lineage during acute colitis in WT mice but not in *IFN-γR KO* mice (Supplementary Fig. 1a). We did not observe changes in the enteroendocrine cell lineages (Supplementary Fig. 1b). These data suggest that IFN-γ promotes an enrichment of KRT20-expressing cells upon injury.

### IFN-γ kills colonocytes and drives progenitor differentiation
To explore these effects in more detail, we analyzed single-cell RNA sequencing (sc-RNAseq) data of DSS colitis mice[25]. This revealed that all epithelial cells respond, but the strongest IFN-γ response occurs in enterocytes (Fig. 1g). To test whether the epithelial changes are directly mediated by IFN-γ, we took advantage of organoid cultures,

which enable analysis of direct epithelial responses. We derived colon organoids from untreated WT mice and cultured them either in full medium (FM, medium containing the full complement of growth factors), which enriches for Ki67+ stem and progenitor cells, or in differentiation medium (Diff., growth factor-reduced medium), which enriches for KRT20+ colonocytes. These conditions mimic the in vivo proliferative crypt base compartment or the surface compartment, respectively (Supplementary Fig. 1c). qPCR for stem cell *(Axin2, Lgr5)* and colonocyte *(Krt20, Bmp2)* marker genes confirmed the differentiation states in the two medium conditions (Supplementary Fig. 1d). To directly explore the cellular responses induced by IFN-γ, we performed live cell imaging of organoids cultured in either full or differentiation medium conditions, with or without simultaneous IFN-γ treatment (Fig. 1h and Supplementary Movie 1). Organoids grown in FM and treated with IFN-γ showed reduced growth and more pronounced cell polarity like differentiated colonocytes, as indicated by actin/phalloidin staining (Supplementary Fig. 1e), and upon prolonged treatment a reduction in size (Supplementary Fig. 1f). Immunofluorescence labeling for the differentiation marker KRT20 indicated that IFN-γ-treated progenitor cells gain features of colonocytes (Fig. 1i), which was confirmed by qPCR (Fig. 1j). Those findings hint at an accelerated differentiation of progenitor cells upon IFN-γ treatment.

Based on the in vivo finding that DSS injury enriches KRT20+ cells in an IFN-γ-dependent manner, we expected that KRT20+ cell-enriched organoids would be more resistant to IFN-γ. However, when we examined the effect of IFN-γ treatment on differentiated organoids, we were surprised to find that they rapidly began to disintegrate (Fig. 1h, Supplementary Fig. 1f, and Supplementary Movie 1). A stronger reduction in cell viability of differentiated cells upon IFN-γ treatment was further confirmed by CellTiter-Glo® Luminescent Cell Viability Assay (Supplementary Fig. 1g). TUNEL-staining for DNA-fragments of apoptotic cells in colon sections from DSS-treated WT mice confirmed that apoptotic cells were found at the surface of the crypts, where mature colonocytes are located, which was further validated by co-staining for KRT20+ cells (Supplementary Fig. 1h). H&E staining also showed a DSS-induced partial disintegration of the colonic epithelial surface in WT mice that was less pronounced in IFN-γ KO mice, indicating that the loss of mature colonocytes is largely dependent on the effects of IFN-γ (Supplementary Fig. 1i). To explore if apoptosis of surface colonocytes was stronger at an earlier time point of DSS-induced colitis, we stained for cleaved caspase 3 (C-CASP3) and indeed found an extrusion of apoptotic cells already on day 3 of DSS treatment (Supplementary Fig. 1j). However, the overall number of apoptotic cells in vivo was rather low, likely because they are quickly extruded and moved away with the feces through peristalsis. Together, our data suggest that IFN-γ has a widespread direct effect on the colonic epithelium: a reduction of mature colonocytes and de-novo differentiation of progenitors towards the colonocyte lineage.

### IFN-γ-induced KRT20+ colonocytes differ from homeostatic
To elucidate the effects of IFN-γ on differentiation, we performed sc-RNAseq of organoids cultured in FM treated with IFN-γ compared to untreated controls (Supplementary Fig. 2a). We clustered the organoid cells by cell type-specific characteristics, which confirmed that most cells in the FM condition are stem cells and proliferating progenitor cells (Supplementary Fig. 2b, red and green cell populations). We visualized the expression of marker genes for stem cells, transit-amplifying cells, colonocytes, and secretory cells in untreated and IFN-γ-treated conditions (Supplementary Fig. 2b, lower panel). We observed a strong reduction in stem and secretory cell identity in response to IFN-γ, consistent with the in vivo observations and our previous findings in human organoids[21]. This finding was confirmed by RT-PCR for secretory cell markers (Supplementary Fig. 2c) and for the

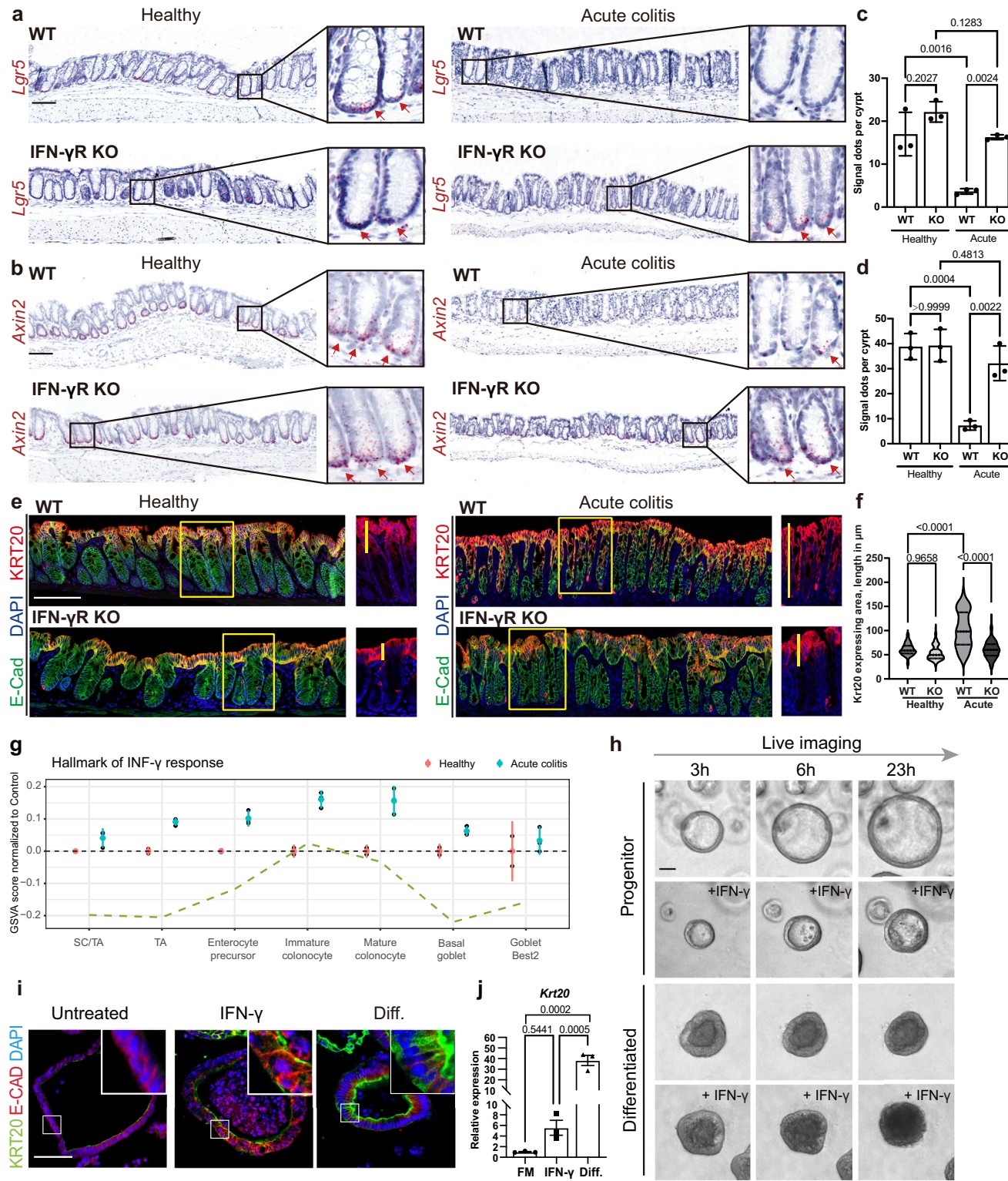

stem cell marker *Lgr5* and the WNT target gene *Axin2*, which were both significantly downregulated upon IFN-γ treatment to levels similar to those observed in differentiated organoids (Supplementary Fig. 2d). In contrast, the sc-RNAseq data revealed a divergent expression of colonocyte markers upon IFN-γ treatment. In response to IFN-γ, the expression of the surface colonocyte marker *Bmp2* decreased (Supplementary Fig. 2f and Supplementary Table 1). However, IFN-γ-treated organoids did not approach the expression levels of differentiation markers observed in fully differentiated organoids and co-expressed markers of transit-amplifying cells (Supplementary Fig. 2b). We

extracted the differentially regulated genes of IFN-γ-treated KRT20+ colonocytes, which revealed changes of colonocyte-related genes like Bmp2 and Aqp8, as well as IFN-γ and inflammatory-related genes like Cd74, Cxcl10, and Ly6e (Supplementary Data 1). To explore the proliferative activity of IFN-γ-treated organoids, we performed immunofluorescence analysis for the pan-proliferation marker Ki67, which showed decreased proliferation upon IFN-γ exposure (Supplementary Fig. 2g, i). Since IFN-γ leads to increased expression of a subset of proliferative markers, although overall proliferation is in fact decreased, we explored whether these cells show signs of cell-cycle

**Fig. 1 | IFN-γ-induced lineage replacement of colonic epithelia upon injury.**
**a** Single-molecule-ISH (sm-ISH) (×200 magnification) was used to assess the stem cell marker *Lgr5* in healthy and acute colitis (5 days DSS treatment) colon tissue of WT and *IFN-γR KO* mice. **b** *Axin2*, a Wnt-target gene, was evaluated using sm-ISH as in (**a**). Scale bar: 100 μm. **c, d** Quantification of *Lgr5* (**c**) and *Axin2* (**d**) expression. *n* = 3 mice per group. **e** Immunofluorescence (IF) (×600 magnification) was employed to compare expression of the differentiated colonocyte marker KRT20 in healthy and acute colitis colon tissue sections of WT and *IFN-γR KO* mice. The length of KRT20-expressing crypts was quantified (**f**). Scale bar: 100 μm. *n* = 3 with 45 measured crypts each. Data are represented as mean ± SD. **g** IFN-γ activation by cell population upon DSS (acute colitis) in vivo, as measured by GSVA enrichment score of the level above the average level in control (healthy) mice from an online sc-RNAseq dataset (GSE201723). The dashed line indicates the average control level for each population. SC stem cells. TA transit-amplifying cells. *n* = 3 mice per group. Error bars indicate ±2 SEM. **h** Live-cell imaging (×200 magnification) was employed to demonstrate IFN-γ-driven cell disintegration in differentiated organoids, comparing proliferative progenitors and differentiated colonocytes with or without IFN-γ treatment for 23 h. Scale bar: 50 μm. **i** IF staining (×600 magnification) of KRT20 in progenitors, IFN-γ-treated progenitors, and differentiated organoids (Diff.). Scale bar: 60 μm. The experiments were repeated independently three times with similar results in (**h**) and (**i**). **j** mRNA expression of *Krt20*, assessed by RT-PCR in organoids grown in FM with or without IFN-γ treatment or in differentiation medium; *n* = 3 biological replicates. All *p*-values above were determined using one-way ANOVA (two-sided) with no adjustments for multiple comparisons. Unless otherwise indicated, data are represented as mean ± SD. Source data is provided as a Source Data file.

arrest. Indeed, cells treated with IFN-γ showed greatly elevated levels of the cell cycle arrest marker p21 (Supplementary Fig. 2h, j).

## Colon assembloids reveal IFN-γ-driven colonocyte replacement

To explore cell type-specific responses to IFN-γ simultaneously in different colon epithelial cell states within an environment that resembles the native crypt structure, we used our recently established colon assembloids, which consist of both stromal and epithelial cells and contain both stem and differentiated cells in an in vivo-like crypt model system[2]. Assembloids resemble in vivo-like crypts with homeostatic organization and cell turnover, composed of proliferative Ki67+ cells at the crypt base and differentiated KRT20+ cells at the surface[2]. The system requires only minimal exogenous growth factors since the key niche factors are produced by the stromal compartment itself. Treating assembloids with IFN-γ altered the compartmentalization and caused flattening of the crypt-like structures (Fig. 2a), which coincides with a decreased complexity of the KRT20+ compartment (Fig. 2b). KRT20 expression appeared to show a different pattern within the assembloids following IFN-γ treatment. Whereas in control colonocytes, KRT20 expression appears to be evenly distributed throughout the cells, in IFN-γ-treated assembloids, KRT20 expression is more concentrated on the apical side of the colonocytes. (Fig. 2b). However, the overall proportion of KRT20+ cells was not changed (overview image: Extended Fig. 3a, and quantification Fig. 2c). Of note, IFN-γ treatment increased the number of KRT20+ cells that were also positive for the proliferation marker Ki67+, as assessed by analysis of multiplex-immunofluorescence staining (Fig. 2b see channel composition on the right, and quantification in Fig. 2d). At the sites of crypt flattening, we also observed high numbers of apoptotic cells (labeled in green) in the assembloid lumen (Fig. 2b, e). Thus, we hypothesized that IFN-γ accelerates the death of "old" KRT20+ cells, while promoting the appearance of a "new" Ki67+/KRT20+ double-positive cell population. To further explore the rapid cell-type-specific effects of IFN-γ, we performed short-time lineage tracing in assembloids. We constructed hybrid assembloids with WT stromal cells and epithelial cells derived from either *Axin2CreErt2/Rosa26-tdTomato* or *Krt20CreErt2/Rosa26-tdTomato* mice and performed lineage tracing in culture induced by 4-OH-tamoxifen (4-OHT) (Fig. 2f–k). In these systems, the tamoxifen-inducible Cre recombinase is either expressed under control of the stem and progenitor marker gene *Axin2*, or under control of the differentiation marker gene *Krt20*. 4-OH-tamoxifen shuttles the Cre-recombinase to the nucleus to remove the transcriptional stop cassette and enable expression of the red fluorescent protein tdTomato. Consequently, cells that derive from either Axin2 or Krt20 expressing cells can be detected via detecting the tdTomato fluorophore with fluorescent microscopy of whole-mounts co-stained with actin/phalloidin to visualize the cell boundaries (Fig. 2g) or on sections (Fig.2i, k). We induced the tracing 72 h after assembly of the assembloids by adding 4-OHT to the cultures for 24 h. 24 h later, we started IFN-γ treatment for 24 h (Fig. 2f). We chose this tracing approach to label the KRT20+ cell population that was present prior to IFN-γ

treatment. In the controls, KRT20+ colonocyte lineage traced, fluorescently labeled cells (green) were present in the upper part of the crypt unit (Fig. 2g, left). In contrast, IFN-γ treatment resulted in a strong reduction of labeled cells (Fig. 2g, right, see further images of KRT20+ lineage traced cells in assembloids in Supplementary Fig. 3b). Quantification of the KRT20+ lineage-traced cells revealed a drop from 38% to 13% KRT20+ traced cells in the remaining epithelium (percentage of tdTomato+ cells from all DAPI+ epithelial cells in field of view, Fig. 2g). Similarly, sc-RNAseq analysis of colonocytes during acute colitis revealed a marked reduction of mature colonocytes while precursor and immature colonocytes remained unchanged (Supplementary Fig. 3c, d). To test for the accelerated generation of daughter cells from the Axin2+ progenitor population upon IFN-γ treatment, we traced the Axin2+ cells. Lineage tracing of the crypt base *Axin2*-expressing cell population revealed an increase of labeled epithelial cells by up to 50% upon IFN-γ treatment compared to untreated assembloids (Fig. 2h). Immunofluorescence for KRT20 of these *Axin2*-tdTomato-traced assembloids indeed revealed that IFN-γ treatment increased the number of KRT20-expressing cells that co-express tdTomato and thus derive from the Axin2+ compartment (Fig. 2i–j). To further explore whether proliferating progenitor cells give rise to Krt20+ cells or acquire differentiation features upon IFN-γ treatment, we performed EdU-labeling in assembloids to label progenitor cells in S-phase and co-stained them with KRT20 (Supplementary Fig. 3e). Two hours before IFN-γ treatment, the base-analog EdU was administered to label S-phase cells. After 24 h of culture, control assembloids exhibited EdU+ cells in Krt20-negative crypt-base cells and only rarely in Krt20+ cells. In contrast, upon IFN-γ treatment, EdU+ labeling was detected more frequently in KRT20-expressing cells (Supplementary Fig. 3e). To monitor the fate of differentiated KRT20+ cells upon IFN-γ treatment, we further analyzed Krt20CreEr-tdTomato labeled cells in assembloids. We first induced lineage tracing, which was followed by IFN-γ treatment (Fig. 2k). After 24 h of IFN-γ treatment, most traced cells had been shed into the lumen and expressed the apoptotic marker C-CASP3 (Fig. 2k). In summary, the lineage tracing experiments in assembloids revealed that IFN-γ treatment fosters extrusion of terminally differentiated KRT20+ cells while simultaneously promoting differentiation of Axin2+ progenitors into Krt20+ colonocyte lineage. The data confirm the differential response of proliferative progenitors and differentiated colonocytes to IFN-γ and provide evidence for colonocyte "lineage replacement.": while the already differentiated colonocytes are extruded from the epithelial sheet, IFN-γ promotes the generation of new KRT20+ cells that are directly derived from proliferative crypt-base cells.

## IFN-γ mediates BMP-2-producing colonocyte loss in colitis

The organoid and assembloid data revealed an IFN-γ-dependent reduction of terminally differentiated KRT20+ cells and the presence of at least partially differentiated Krt20+ colonocytes that differed in their expression of other markers, such as BMP2 (Supplementary Fig. 2b). To explore whether IFN-γ has similar effects in vivo and whether

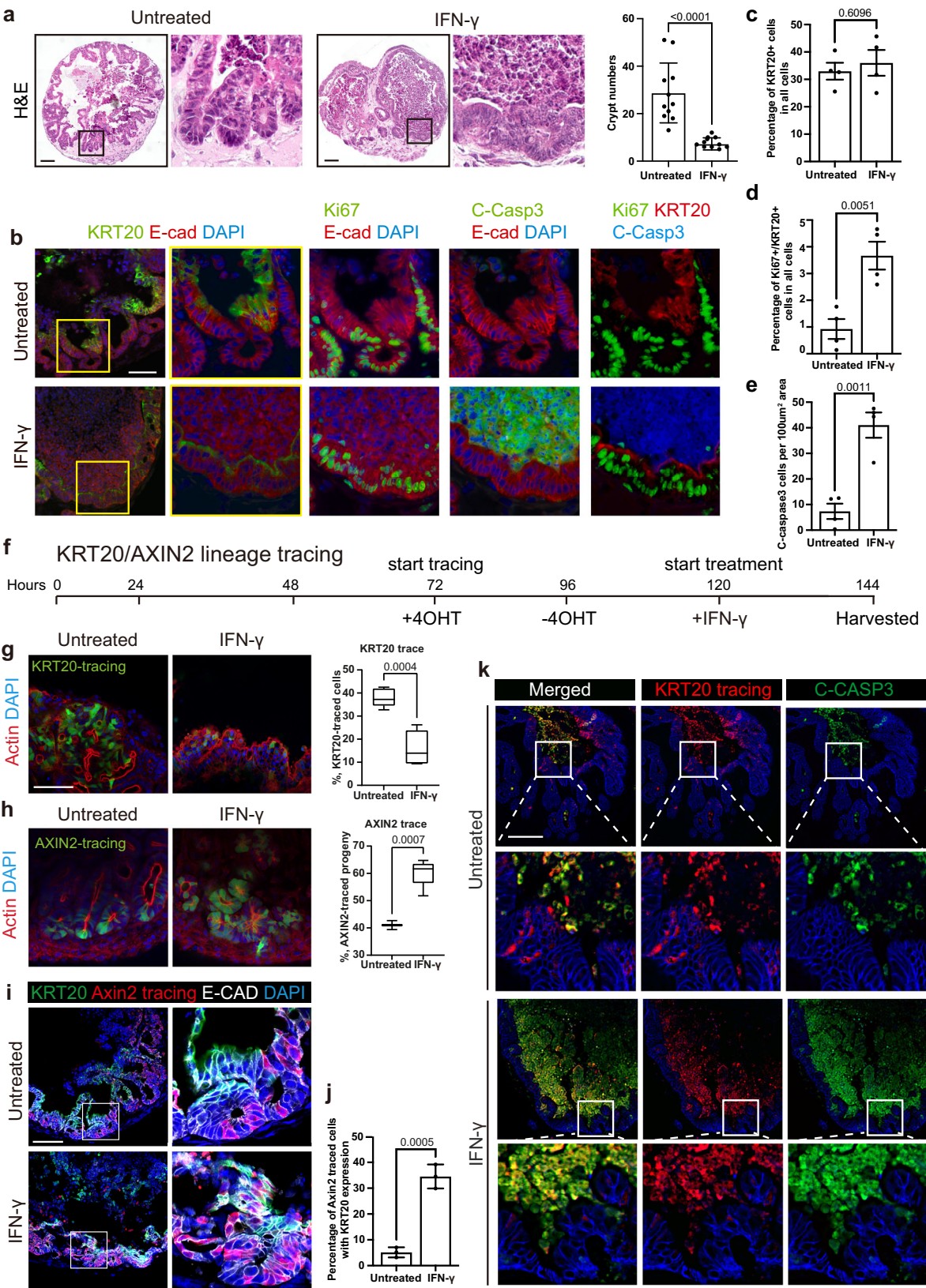

they are connected to regeneration, we analyzed changes in colon tissue from WT and IFN-γR KO mice at the acute colitis stage. At homeostasis in vivo, surface colonocytes expressed the differentiation marker Bmp2, as shown by ISH (Fig. 3a, left). In contrast, during acute colitis, Bmp2 expression was strongly reduced in WT but not in IFN-γR KO mice, as was the BMP target gene Id1 (Fig. 3a, b right). Indeed, sc-

RNAseq analysis illustrated a marked decrease in the BMP signaling status in colonocytes during acute colitis (Supplementary Fig. 4a). These data suggest that IFN-γ is involved in the reduction of Bmp2-expressing colonocytes and in the enrichment of de-novo differentiated colonocytes that neither express Bmp2 nor have active BMP signaling. This may offer a mechanism that selects colonocytes for

**Fig. 2 | Assembloids confirm the effects of IFN-γ on exfoliating KRT20-traced mature colonocytes and promoting differentiation of progenitors into a new KRT20+ lineage. a** H&E staining (×200 magnification) demonstrating the effect of IFN-γ on crypt formation in assembloids, with quantification on the right. Scale bar: 100 μm. *n* = 11 from three biological replicates. **b** KRT20, Ki67, and cleaved-caspase 3 were evaluated via multiplex IF staining (×200 magnification) in assembloids treated with or without IFN-γ. Channel composition on the right. Quantifications are shown in (**c**–**e**). Scale bar: 100 μm. *n* = 4 biological replicates. **f** Scheme of KRT20/AXIN2 lineage tracing. **g**, **h** Confocal microscopy images (×600 magnification) of whole-mount staining of assembloids co-stained with actin/phalloidin derived from **g** *Krt20CreERT2/Rosa26-tdTomato* mice or **h** *Axin2CreERT2/Rosa26-tdTomato* mice. KRT20 (**g**) or AXIN2 (**h**) tracing was induced by 800 nM tamoxifen to label the differentiated lineages (**g**) or progenitor lineages (**h**) followed by treatment with or without IFN-γ for 24 h, with quantifications on the right. The box represents the interquartile range (IQR), bounded by the 25th percentile (lower edge) and the 75th percentile (upper edge). The center of the distribution is indicated by the median (solid line within the box). The minimum and maximum values are marked by the whiskers, extending from the lower and upper edges of the box, respectively. *n* = 5 individual assembloids. Scale bar: 100 μm. **i** Co-staining of Krt20 and RFP (×600 magnification) was performed in AXIN2-traced untreated or IFN-γ-treated assembloids derived from *Axin2CreERT2/Rosa26-tdTomato* mice. The percentage of AXIN2-traced KRT20+ cells in all KRT20+ cells was quantified (**j**). *n* = 3. Scale bar: 100 μm. **k** Immunofluorescence staining (×200 magnification) of RFP, C-CASP3, and E-cadherin was performed in assembloids derived from *Krt20CreERT2/Rosa26-tdTomato* mice. KRT20 tracing was induced by 800 nM tamoxifen to label the differentiated lineage, followed by treatment with or without IFN-γ for 24 h. Scale bar: 100 μm. The experiment was repeated independently three times with similar results. The scheme of KRT20 lineage tracing is presented in the upper panel. All *p*-values above were determined by Student's *t*-test (two-sided). All data are represented as mean ± SD. See also Supplementary Fig. 3. Source data are provided as a Source Data file.

extrusion vs tissue persistence and participation in regeneration. To corroborate this hypothesis, we pretreated colon organoids with BMP-2 followed by IFN-γ treatment and vice versa. BMP-2 treatment induced differentiation of organoids, as did IFN-γ treatment (Fig. 3c). In congruence with our differentiation experiments (see Fig. 1i and Supplementary Fig. 1c), organoids that were pretreated with BMP-2 differentiated and subsequently died upon treatment with IFN-γ (Fig. 3d). In contrast, IFN-γ pretreatment followed by BMP-2 treatment protected cells from apoptotic cell death (Fig. 3e). Indeed, live cell imaging of organoids treated with IFN-γ and cultured with a live-marker for apoptotic cells revealed a selective shedding of apoptotic cells (Supplementary Movie 3 and quantification Supplementary Fig. 4b). RT-PCR revealed that IFN-γ pretreatment changed the BMP responsiveness of organoids—in addition to reducing Bmp2 expression in organoids (Fig. 3f) and assembloids (Supplementary Fig. 4 c). Furthermore, IFN-γ downregulated the BMP-receptor Bmpr1a, as well as the BMP target gene Id1 (Fig. 3f).

## IFN-γ signaling participates in regeneration in vivo

A recent proteomic analysis of the effects of disrupted BMP signaling in the colon revealed increased annexin-A1 (Anxa1) expression[26]. ANXA1 is a phospholipid-binding protein involved in tissue repair[27] and is highly expressed in regenerative epithelium of the colon[13]. To explore if the IFN-γ-driven reduction of BMP signaling also results in altered *Anxa1* expression, we performed ISH for *Anxa1*. *Anxa1* expression was strongly upregulated in all epithelial cells along the crypt axis in WT but not in *IFN-γR KO* mice during acute colitis (Fig. 4a). Since *Anxa1* is strongly expressed in YAP activated intestinal cells[13,15] and YAP signaling is an important pathway controlling regeneration, we analyzed the expression and localization of the active, non-phosphorylated form of YAP. During acute colitis, active YAP was present in the nucleus of epithelial cells along the entire crypt axis in WT but not in *IFN-γR KO* mice (Fig. 4b). Similarly, in colon organoids (which already show high YAP levels at baseline when grown in full medium), nuclear YAP levels remained high upon IFN-γ treatment (Supplementary Fig. 5a, b). This finding was corroborated in the epithelium of assembloids treated with IFN-γ, which also strongly upregulated *Anxa1* (Fig. 4c, d) and exhibited high levels of nuclear YAP (Fig. 4e, f).

A hallmark of regenerative epithelia, apart from *Anxa1* expression and nuclear YAP, is high proliferative activity[13]. However, during acute colitis, proliferative activity was strongly suppressed in WT but not in *IFN-γR KO* mice (Supplementary Fig. 5c). At a later time point, 4 days after the acute colitis stage, a restoration of proliferation was observed (Supplementary Fig. 5c, lower panel). The reverse was true for the cell cycle arrest marker p21, which increased during acute colitis and decreased again during the post-acute stage in WT mice (Supplementary Fig. 5d), while *IFN-γR KO* mice did not show these responses.

Similarly, after IFN-γ treatment, assembloids also showed an abundance of nuclear YAP in cells with decreased Ki67 expression and increased KRT20 expression (Supplementary Fig. 5e). To corroborate this finding in vivo, we performed multiplex-immunofluorescence analysis for KRT20, Ki67, P21, and non-P-YAP of tissue sections from untreated and DSS-treated IFN-γ KO and WT mice. As expected, WT but not IFN-γ KO mice revealed cell cycle-arrested surface colonocytes that express nuclear Yap during acute colitis (Fig. 4g). To explore whether the IFN-γ-induced colonocytes have a higher potential to regenerate compared to normal KRT20+ cells, we compared the reseeding capacity of KRT20+ traced cells (Krt20CreEr-tdTomato) from assembloids that were untreated or treated with IFN-γ. Assembloids were dissociated, and single cells were seeded back into the classical organoid culture (Supplementary Fig. 5f). The data revealed an increased regrowth of KRT20+ traced cells after IFN-γ-treatment. Thus, IFN-γ-induced Krt20+ colonocytes retain a regenerative capacity.

## IFN-γ-induced BMP-2 loss triggers stromal HGF expression

To identify whether and how the IFN-γ-induced colonocytes are stimulated to enter a proliferative regenerative state, we added different factors to IFN-γ-pretreated organoid cultures that are reported to be involved in injury repair. Analysis of published sc-RNAseq data[28] verified that expression of those ligands was elevated in subsets of stromal cells during DSS colitis (Supplementary Fig. 6a). Organoids were pretreated with IFN-γ together with interleukin 6 (IL-6)[29], IL-22[30], IL-33[31], neuregulin 1 (Nrg1)[32] or HGF[33,34] and organoid size quantified after 2 days (Fig. 5a and Supplementary Fig. 6b). The IFN-γ-dependent reduction in proliferation was reversed by HGF, while Nrg1, IL-22, IL-33 or IL-6 was not sufficient to rescue organoid growth (Fig. 5a and Supplementary Fig. 6b). Of note, single-factor treatment without IFN-γ did not affect organoid growth (Supplementary Fig. 6c). Analysis of the published sc-RNAseq data[28] of stromal cells revealed that *Hgf* expression is low at homeostasis (Fig. 5b, left) but strongly increased in different subsets of stromal cells during colitis (Fig. 5b, right). Those stromal cell populations (fibroblasts 1 and 2) that upregulated *Hgf* most strongly also showed the strongest IFN-γ response in the context of colitis based on GSVA analysis (Fig. 5b, c and Supplementary Fig. 6a). ISH of colon tissue sections revealed a marked increase in *Hgf* expression in stromal cells upon DSS treatment in WT mice, which was significantly less prominent in *IFN-γR KO* mice (Fig. 5d, e). To assess whether IFN-γ or loss of BMP-2 directly influences HGF expression, we treated primary stromal cells with IFN-γ or BMP-2. We observed that IFN-γ induced upregulation of *Hgf*, while treatment with BMP-2 resulted in downregulation of *Hgf* (Fig. 5f). Once cells were treated with BMP-2, they did not show an increase in *Hgf* expression upon exposure to IFN-γ, indicating that loss of BMP-2 signaling is a prerequisite for the increased *Hgf* production (Fig. 5f). Likewise, when we first treated

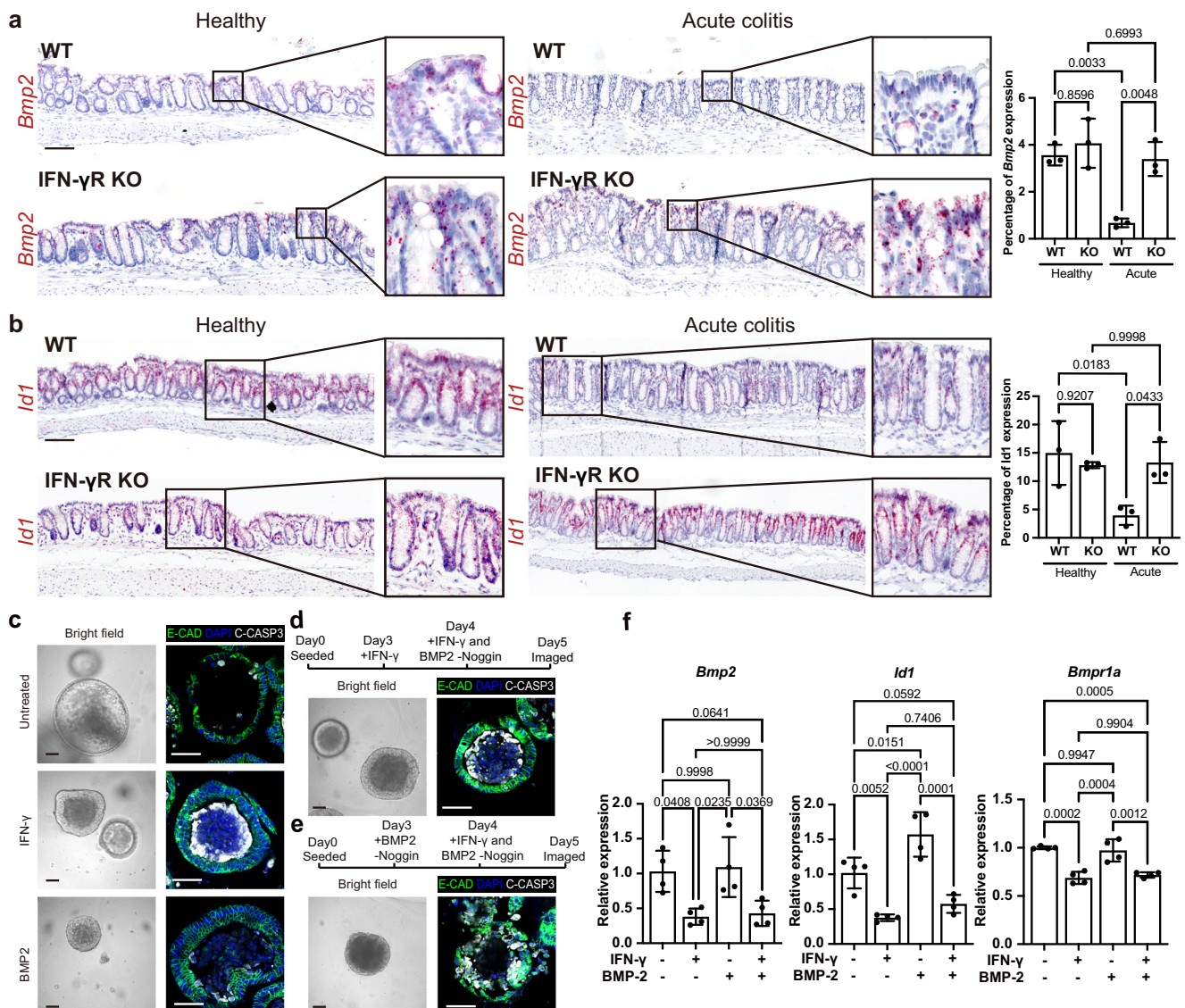

**Fig. 3 | IFN-γ triggers loss of BMP-2 signaling. a** Sm-ISH (×200 magnification) showing the expression of *Bmp2* in healthy and acute colitis colon tissue of WT and *IFN-γR KO* mice. Quantification is shown in the right panel. *n* = 3, scale bar: 100 μm. **b** Expression of *Id1*, a BMP signaling target, was evaluated via sm-ISH (×200 magnification) in healthy and acute colitis colon tissue of WT and *IFN-γR KO* mice. Quantification is shown in the right panel. *n* = 3, scale bar: 100 μm. **c** Progenitor organoids with or without 48 h IFN-γ or BMP-2 treatment. Brightfield (×200 magnification) and C-CASP3 staining images (×600 magnification) are shown. Scale bar in brightfield: 100 μm. Scale bar in IF images: 50 μm. **d, e** Sequential treatment of BMP-2 and IFN-γ was performed on progenitor organoids, with bright field and

C-CASP3 staining images. The scheme is shown above. Scale bar in brightfield: 100 μm. Scale bar in IF images: 50 μm. **f** mRNA expression of *Bmp2*, *Id1*, and *Bmpr1a* were assessed using RT-PCR in untreated, BMP-2 treated, IFN-γ treated, and BMP-2 + IFN-γ treated progenitors. *n* = 4 independent organoid lines from different mice. Experiments displayed in (**c–e**) were repeated independently three times with similar results. Data are represented as mean ± SD. All *p*-values above were determined using one-way ANOVA (two-sided) with no adjustments for multiple comparisons. See also Supplementary Fig. 4. Source data are provided as a Source Data file.

stromal cells with BMP-2 to mimic the homeostatic state, and then removed BMP-2 from the culture to mimic the loss of BMP-2 in the acute injury state, we found that the cells re-gained their responsiveness to IFN-γ and upregulated *Hgf* expression (Fig. 5g). We confirmed the inverse regulation of BMP signaling activity and *Hgf* expression during colitis by sc-RNAseq data analysis (Fig. 5h). Indeed, impeding BMP-signaling by adding the BMP antagonist Noggin resulted in upregulation of HGF in cultured stromal cells following addition of IFN-γ (Fig. 5i). To confirm the regulatory circuitry of BMP signaling and HGF production in stromal cells, we generated *Col1a2Cre/Alk3^fl/fl^* mice, which lack BMP signaling in their stromal compartment, and performed ISH of colon sections from these mice. The genetic loss of BMP receptor1 (Alk3) in stromal cells in vivo resulted in a strong increase in

*Hgf* expression, which was found outside of the crypt base (Fig. 5j). Adjacent epithelial cells outside of the crypt base showed increased proliferative activity, as assessed by Ki67 staining (Fig. 5j, middle panel), while an increase in active nuclear Yap was not observed (Fig. 5j, lower panel). This suggests that the loss of BMP signaling may allow for the remodeling of the stroma and increased production of HGF after injury.

## HGF promotes proliferation in IFN-γ-induced colonocytes

HGF is a growth and motility factor involved in the control of different cellular processes such as survival, growth, angiogenesis, and metastasis[35]. For full functionality, the HGF precursor protein is proteolytically processed to the mature two-chain form of HGF (tc-HGF)[36].

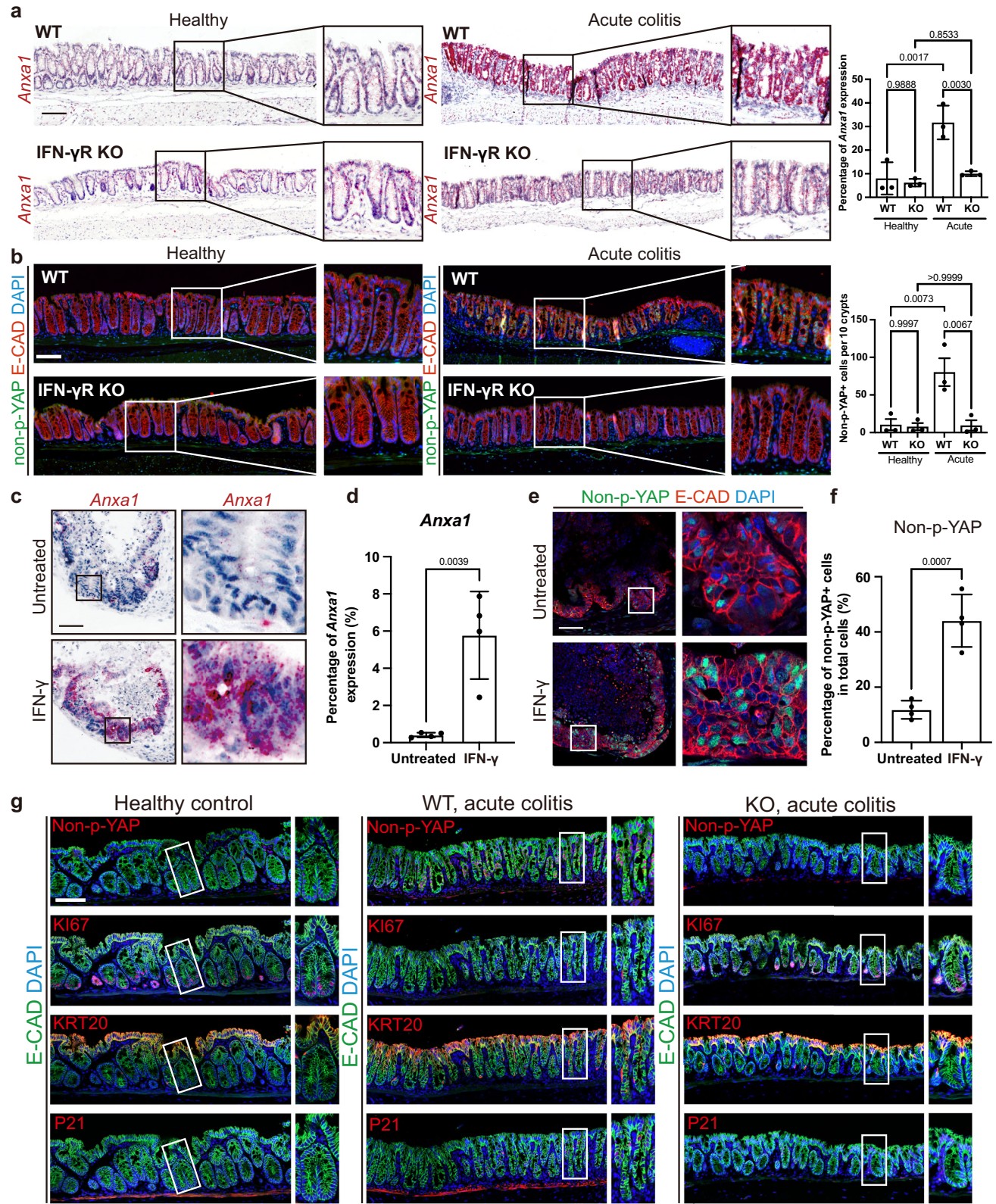

To utilize functional HGF in all our experiments, we produced tc-HGF (from now on referred to as HGF; see Methods) and tested its efficacy in the well-established MDCK-scatter assay[37]. The produced HGF induced epithelial-to-mesenchymal transition of MDCK cells already at low concentration (Supplementary Fig. 6d). To explore the capacity of HGF to reactivate organoid growth, we treated organoids grown in FM with IFN-γ prior to reseeding single cells and adding HGF. As expected,

IFN-γ reduced the reseeding capacity of organoids, but this was alleviated by HGF (Fig. 6a). Of note, HGF was not sufficient to rescue the reseeding capacity of organoids grown in differentiation medium, indicating that HGF specifically targets the pre-regenerative but not the homeostatic colonocytes (Fig. 6a). To further explore the effect of HGF, we performed live cell imaging and followed organoid growth over 24 h, which confirmed the rescue effects (Fig. 6b and

**Fig. 4 | Induction of partial regenerative features by IFN-γ. a** Expression of *Anxa1*, a regenerative marker, shown by sm-ISH (×200 magnification) in healthy and acute colitis colon tissue of WT and *IFN-γR KO* mice, with quantification in the right panel. $n = 3$ mice. ns not significant. Scale bar: 100 μm. **b** Evaluation of non-phosphorylated YAP (non-p-YAP) via IF staining (×200 magnification) in healthy and acute colitis colon tissue of WT and *IFN-γR KO* mice. Quantification in the right panel. $n = 3$ mice. ns not significant. Scale bar: 100 μm. The *p*-values in (**a, b**) were determined by one-way ANOVA (two-sided) with no adjustments for multiple comparisons. **c** Sm-ISH (×400 magnification) evaluation of *Anxa1* in assembloids with or without 24 h IFN-γ treatment, with quantification in (**d**). $n = 4$ independent assembloids. Scale bars: 100 μm. **e** Assessment of non-p-YAP by IF staining (×200 magnification) in assembloids with or without 24 h IFN-γ treatment, with quantification in (**f**). $n = 3$ independent assembloids. Scale bar: 100 μm. The *p*-values in (**d, f**) were determined by Student's *t*-test (two-sided). **g** Multiplex immunofluorescence staining (×600 magnification) of non-p-YAP, E-CAD, KI67, KRT20, and p21 in healthy WT control, *IFN-γR KO* mice with 5 days DSS, and WT mice with 5 days DSS. Experiments were repeated independently three times with similar results. Scale bar: 100 μm. See also Supplementary Fig. 5. All data are represented as mean ± SD. Source data is provided as a Source Data file.

Supplementary Movie 2). Actin/phalloidin staining also revealed that HGF counteracts the IFN-γ-induced cell differentiation (Fig. 6c). We further explored the proliferation status of the organoids by EdU labeling to visualize S-phase cells. While IFN-γ reduced the number of S-phase cells, co-treatment with HGF rescued the number of S-phase cells (Fig. 6d), which was further confirmed by Ki67 staining (Fig. 6e). Moreover, HGF counteracted the IFN-γ-induced increase in p21 staining (Fig. 6f). While HGF treatment rescued cell proliferation and viability in the presence of IFN-γ (Fig. 6g), expression of the stem cell marker genes *Lgr5* and *Axin2* remained low (Supplementary Fig. 7a). HGF also did not revert the IFN-γ-induced changes in expression of the differentiation markers *Krt20* and *Bmp2* or the IFN-γ target genes *Xaf1* and *Cxcl11* (Supplementary Fig. 7b, c). To further explore if HGF promotes the regrowth of IFN-γ-induced colonocytes, we treated *Krt20CreEr-tdTomato* organoids with IFN-γ and explored the colony-forming capacity of the KRT20 lineage. First, we confirmed a higher abundance of KRT20+ cells upon IFN-γ treatment compared to untreated controls (Supplementary Fig. 7d). Single-cell seeding into expansion medium showed a higher regrowth rate of tdTomato+ cells in the HGF-treated condition (Supplementary Fig. 7e), and those KRT20 traced cells expressed the proliferative marker Ki67 (Supplementary Fig. 7f). We explored the expression of IFN-γ receptors in the different conditions and did not observe a significant downregulation, implicating that IFN-γ signaling is bypassed by other mechanisms (Supplementary Fig. 7g). In contrast to the IFN-γ induced colonocytes, HGF was not sufficient to promote survival of differentiated colonocytes exposed to IFN-γ (Fig. 6h, i and Supplementary Fig. 7h, i). In summary, our data reveal that IFN-γ-induced colonocytes retain regenerative capacity and are stimulated to proliferate by stromal-derived HGF.

## Discussion

We demonstrate here that the pro-inflammatory cytokine IFN-γ, produced by immune cells in response to chemically induced colitis, actively reshapes the cellular organization of epithelial crypts and stroma. IFN-γ orchestrates rapid lineage replacement by inducing apoptosis of terminally differentiated BMP-2-expressing colonocytes alongside enrichment of distinctive colonocytes that harbor regenerative capacity. These cells exhibit a strongly reduced expression of BMP-2, which not only leads to increased cellular plasticity in the epithelium but also has an impact on the surrounding stromal cells. Reduction of BMP signaling results in increased HGF expression, which in turn causes proliferation of IFN-γ-induced colonocytes. Effectively, this regulatory mechanism enables epithelial reprogramming into the regenerative state from an IFN-γ-induced colonocyte state and explains how non-stem cells contribute to regeneration. In this study, we used two different primary cell culture systems—organoids and assembloids. The classical organoid culture system has the advantage of providing insights into pure epithelial responses and allowing the performance of live imaging, while the assembloid system has the advantage of revealing epithelial responses in a setting that closely resembles the structure and cell type composition of the tissue in vivo.

While homeostatic epithelial turnover is organized in a hier-archical manner, this hierarchy is lost upon injury: it has been observed

that various intestinal epithelial cell types can enter the proliferative regenerative state and contribute to epithelial regeneration[7–12]. The organizing signals that drive the regenerative state are distinct from those of homeostatic cellular turnover. For example, regenerative cells do not rely on high WNT signaling but are instead characterized by YAP activity[13]. Although this state is well described in the small intestine and colon, the mechanisms that reprogram the epithelium are not yet fully understood. Our data here reveal that it is an orchestrated endogenous process that extends to undamaged tissue regions, involving the interplay between stromal and epithelial cells and is actively driven by inflammatory IFN-γ signaling.

Epithelial plasticity is considered to be essential for injury repair, and various studies have demonstrated Lgr5+ cell-independent crypt regeneration upon injury. While originally, label-retaining "reserve" stem cells were proposed to be activated upon Lgr5+ cell loss[38–40], recent data have put a focus on early progenitors as drivers of regeneration. In particular, Shivdasani and collaborators showed that regeneration occurs from early Lgr5+ cell-derived daughter cells of the secretory and absorptive lineages[41]. Several groups have addressed the plasticity of such progenitors using lineage tracing, demonstrating the ability of various lineages, including KRT20+ colonocytes, to replenish lost stem cells upon epithelial damage[10,42–44]. Our data reveal a KRT20+ cell population that is induced and enriched by IFN-γ upon injury. To analyze this, we used the classical colon organoid and the assembloid systems. The latter allows the simultaneous study of different cell states, in particular Krt20+ cells and proliferative cells, whereas the classical colon organoids can be enriched for specific individual cell types. Using these different model systems, we revealed that the IFN-γ-induced KRT20+ colonocytes are distinct from homeostatic, termin-ally differentiated KRT20+ colonocytes, harbor regenerative capacity, and can regain proliferative activity. These IFN-γ-induced KRT20+ cells are characterized by reduced expression of BMP-2 and reduced BMP responsiveness. Further, they are cell cycle arrested while exhibiting nuclear YAP. Nuclear YAP is a hallmark of the regenerative cell state[13] and appears inconsistent with the IFN-γ-induced cell cycle arrest. However, our finding correlates with a previous report by Nava et al. of activated nuclear β-catenin in IFN-γ-treated and cell cycle-arrested intestinal cells[45]. They suggested that IFN-γ induces a biphasic pro-liferative and apoptotic response in intestinal epithelial cells with an initial increased proliferation followed by loss of proliferative activity and apoptosis despite continued β-catenin activation[45]. Together with our data, this implies that IFN-γ shifts the cells towards a colonocyte state that preserves the capacity for regeneration. Here, we also show that the transition from the IFN-γ-induced arrested colonocyte state into the proliferative regenerative cell state involves stroma-derived HGF.

Stromal cells are considered important determinants of tissue homeostasis and form the stem cell niche (reviewed in ref. 46). Importantly, injury is followed not only by reprogramming of the epithelium but also by changes in the stroma, which increase the expression of regenerative factors such as HGF[28]. The mechanisms by which stromal cells respond to injury have not yet been explored in detail. We previously reported that loss of BMP-2-producing epithelial cells and, therefore, the loss of BMP signaling in the surrounding

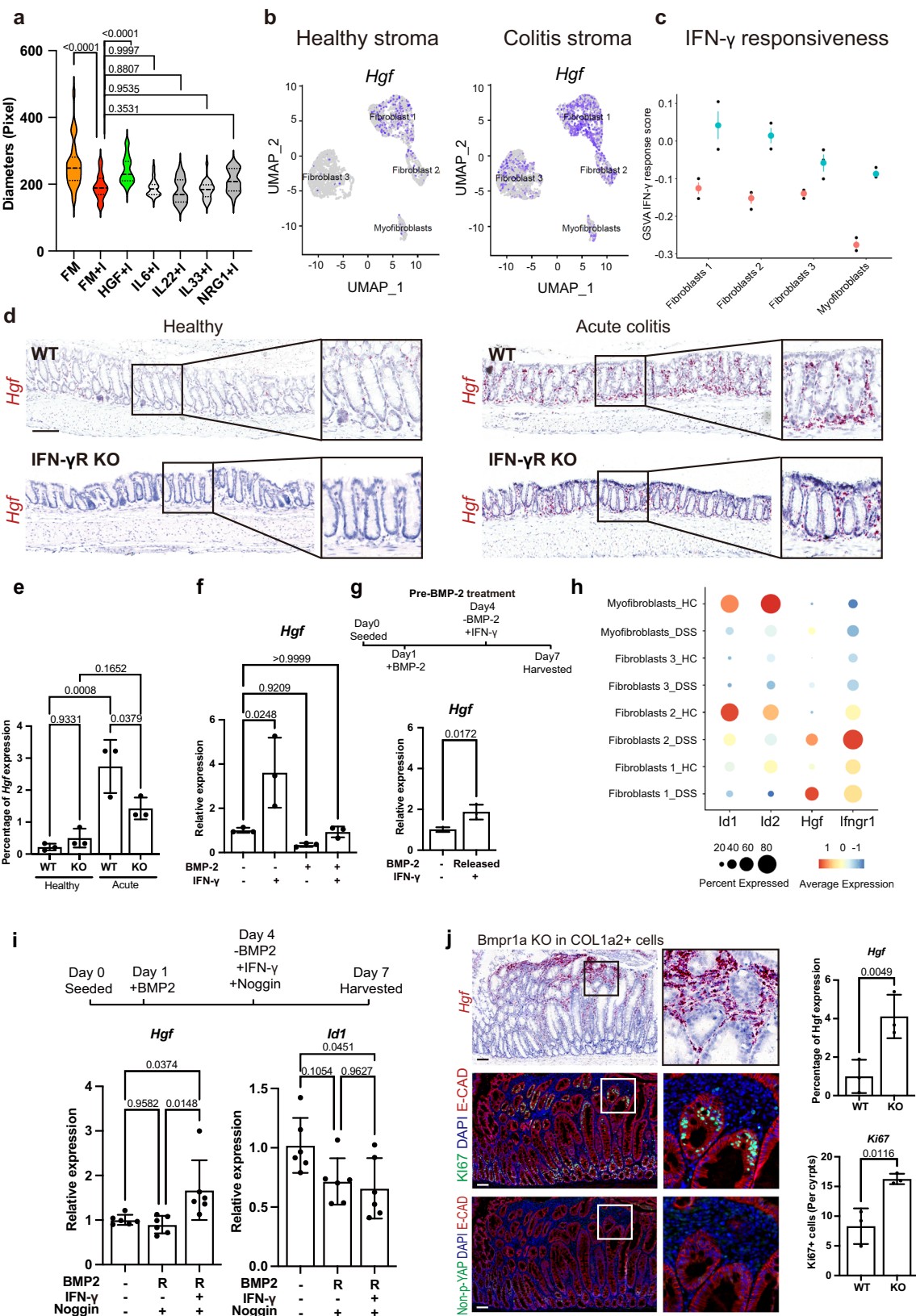

stroma is a critical driver of stromal remodeling[6]. Leedham and collaborators analyzed the role of BMP signaling in intestinal regeneration and showed a reduction of epithelial BMP-4 production and an increase in stromal GREM-1 production during colitis. They demonstrated that the suppression of BMP signaling by stromal upregulation of the BMP antagonist GREM-1 protected from ulcerative lesions upon DSS-mediated colitis induction. The data highlight that suppression of BMP signaling is critical for intestinal regeneration[47].

The present data show that BMP-2 acts as an inhibitor of HGF production in the stroma and that its loss releases the inhibitory effect. The role of BMP in terminal differentiation of epithelial cells is well known[3–5,48]. However, recent reports have identified that BMP signaling

**Fig. 5 | Loss of BMP-2 expression enhances IFN-γ-driven HGF expression.**
**a** Diameter of progenitor organoids treated with growth factors (HGF, IL-6, IL-22, IL-33, and NRG1) in combination with IFN-γ for 48 h. n = 30 organoids from three biological replicates. **b** UMAP expression plots of *Hgf* in stromal cells derived from healthy and acute colitis colon tissue obtained from an online data set (GSE114374). Cells are colored based on the normalized expression of the indicated marker genes. **c** GSVA Analysis of IFN-γ responsiveness of the respective stromal cells obtained from the same online data set as in (**b**) (GSE114374) (red = untreated, blue = IFN-γ treated). n = 3 mice. **d** Evaluation of *Hgf* expression via sm-ISH (×200 magnification) in healthy and acute colitis colon tissue of WT and *IFN-γR KO* mice, followed by quantification (**e**). n = 3 mice. Scale bar: 100 µm. **f** qPCR analysis of *Hgf* expression in primary stromal cells treated with IFN-γ, BMP-2, or both. n = 3 independent organoid lines. **g** qPCR measurement of *Hgf* mRNA in stromal cells after sequential treatment with BMP-2 and IFN-γ, mimicking BMP-2-expressing cells and

IFN-γ secretion in colitis. Schematic representations of the treatment are shown above. n = 3 independent organoid lines. **h** Dot-plot analysis of single-cell RNA sequencing data demonstrating the inverse regulation of *Hgf* expression and BMP-signaling activity in the stroma cell population in colitis conditions. **i** The BMP antagonist Noggin was used to rescue the effect of BMP-2 in primary stromal cells treated with IFN-γ or BMP-2 or both. R: BMP-2 removed from medium. *Hgf* and *Id1* were evaluated via qPCR. n = 6 independent organoid lines. **j** Sm-ISH (×200 magnification) evaluation of *Hgf* and immunofluorescence staining of non-p-YAP and KI67 in *Col1a2Cre/Alk3^{fl/fl}* mice 2 months after tamoxifen injection. Scale bar: 100 µm. Quantifications are shown in the right panel. n = 3 mice. All data are represented as mean ± SD. The *p*-values in (**a**, **e**, **f**, and **i**) were determined using one-way ANOVA (two-sided) with no adjustments for multiple comparisons. The *p*-values in (**g**, **j**) were determined using unpaired Student's *t*-tests (two-sided). See also Supplementary Fig. 6. Source data are provided as a Source Data file.

also impacts the stroma, e.g., by reprogramming trophocytes to switch off expression of pro-proliferative factors[2,49]. Accordingly, our data also show that HGF expression is inhibited by BMP signaling. Furthermore, we reveal that BMP signaling is responsible for a differential responsiveness of stromal cells to inflammatory stimuli in the colon.

Different growth factor systems, such as IL-22[50] and IL-33[51], also contribute to intestinal regeneration; however, we could show that the known "healing" factor HGF[32,33,44] promotes proliferation of IFN-γ-induced colonocytes. At homeostasis, HGF production in intestinal stromal cells is suppressed by epithelium-derived BMP signaling. Thus, the IFN-γ-induced lineage replacement from BMP-high to BMP-low colonocytes during colitis triggers the expression of HGF in stromal cells. HGF signals back to the epithelium to activate proliferation of the BMP-low colonocytes. Our observation is in strong agreement with data from other groups, which in essence demonstrate that HGF signaling controls the exit from the quiescent cell state and induces cell cycle progression and cell division by controlling cell cycle transition from G0 to G1, as well as the entry into and progression through S phase[35,52,53]. The mechanism by which HGF interferes with IFN-γ remains unresolved. However, our data pinpoint the IFN-γ, BMP, and HGF signaling circuitry between epithelium and stroma as crucial for controlling epithelial extrusion and regeneration.

Intestinal cell turnover can be viewed as a mechanism of tissue defense, with cells that reach the surface constantly being shed into the lumen. At homeostasis, the continued cell shedding is compensated for by the production of new cells. In intestinal diseases, the turnover can become unbalanced, and an accelerated rate of epithelial cell shedding can exceed the regenerative capacity of the crypts. For instance, the pathology and disease state of IBD are influenced by the amount of cell shedding[54]. Here, we show that in the context of injury, cell shedding can be promoted by IFN-γ, which triggers increased exfoliation of terminally differentiated cells.

The intestinal cell turnover rate of germ-free mice is much lower than that of conventionally housed mice[55], and infection increases cellular turnover to promote clearance of pathogens or parasites, as shown for the epithelia of the colon[56] and bladder[57] (reviewed in ref. 58). In all those experiments, a participation of IFN-γ can be assumed. Similarly, reduced expulsion of *Salmonella* Typhimurium-infected cells from the intestinal epithelium has been observed in infected IFN-γ KO mice. The authors hypothesized that this was caused by reduced goblet cell function[59]. Our data imply that IFN-γ fosters the extrusion of potentially damaged surface epithelial cells by selectively inducing cell death and exfoliation of differentiated cells to accelerate tissue clearance.

Pathogens infect cells by utilizing cell surface molecules like ACE2 and DPP4, which are highly expressed on differentiated cells[21,60,61]. We thus hypothesize that the accelerated exfoliation of surface cells is not merely a "necessary evil" to release the epithelium from the influence of BMP-2 in readiness for regeneration of damaged areas, but that it also acts as a first defense mechanism. Cell death as an active defense

has been described in enterovirus-infected intestinal epithelial cells, which are actively extruded to protect against the spread of the virus[62]. Thus, the epithelial response to IFN-γ may have evolved to remove any potentially damaged and infected cells from the entire epithelial sheet in order to protect the integrity of the tissue, followed by rapid regeneration.

## Methods

### Mice

All procedures involving animals were approved by the institutional, local, and national legal authorities at the Charité and the Max Planck Institute for Infection Biology (LAGeSo, Berlin, Reg. G0084/17). Mice were bred in pathogen-free conditions, and care and use of animals complied with the European and national regulations, published in the Official Journal of the European Union L 276/33, 22 September 2010. The mice were bred on a 12-h light/12-h dark cycle in a controlled temperature (22.5 ± 2.5 °C) and humidity (50 ± 5%) environment. Both sexes of mice aged from 6 weeks to 12 weeks were used in this study. The *IFN-γR KO* mouse strain (homozygous for the null mutation of the *IFN-γR* gene) has been previously described[6,63] and colitis was induced in alignment with described protocols[12,23] by the administration of 1.5% DSS in drinking water for five consecutive days in *IFN-γR KO* mice (n = 3; 8 weeks) and their WT littermates (n = 3; 8 weeks). The healthy controls for *IFN-γR KO* mice (n = 3; 8 weeks) and their WT littermates (n = 3; 8 weeks) were provided with regular drinking water for the same duration. Mice were sacrificed at the indicated time points to harvest tissue samples for histology. Tissue for organoids and assembloids was collected from the mouse strains *C57bl/6* (Jackson Laboratories) and *Krt20CreErt2/Rosa26-tdTomato* and *Axin2CreErt2/Rosa26-tdTomato* described previously[12]. *Alk3^{fl/fl}* mice were obtained from the laboratory of Yuji Mishina[64]. *Col1a2CreErt2* mice were obtained from the laboratory of Ahmed Hegazy from the Charité[65]. To generate conditional KO mice with depletion of Bmpr1a in COL1A2+ cells, we bred *Alk3^{fl/fl}* mice to *Col1a2CreERT2* mice to get *Col1a2CreERT2/Alk3^{fl/fl}* mice. The *Col1a2CreERT2/Alk3^{fl/fl}* mice (n = 3; 8 weeks) and their WT littermates (n = 3; 8 weeks) were injected with tamoxifen intraperitoneally on three consecutive days to deplete Bmpr1a from Col1a2+ cells. Mice were euthanized two months after tamoxifen injection, and tissue samples were collected for histological analysis.

### Organoids

Organoids were grown based on previously developed protocols[21]. In brief: dissected mouse colon was opened longitudinally, washed, and cut into 2–5 mm long pieces. Five cycles of washes in ice-cold PBS were followed by incubation for 5 min in 10 mM EDTA/PBS at RT. After removal of the buffer, the tissue was incubated on a rotating shaker for 90 min at 4 °C in 2 mM EDTA/PBS supplemented with 2.5 µM DTT. Tissue pieces were transferred to 15 ml PBS and shaken vigorously ten times to separate crypts from the tissue. This procedure was repeated three times, and the crypt-containing buffer was collected and pooled

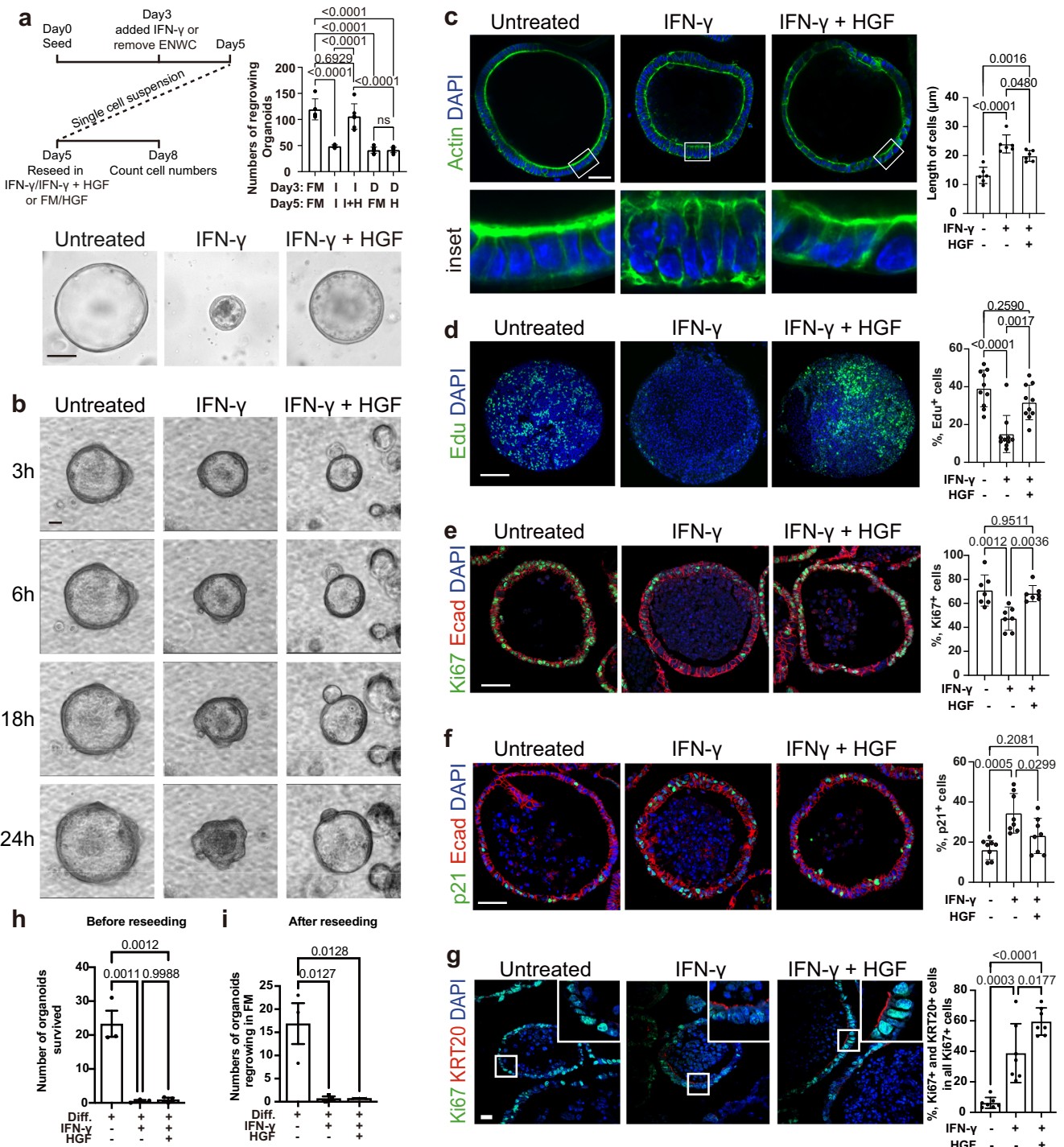

**Fig. 6 | HGF releases the cell cycle arrest imposed by IFN-γ. a** Bright-field images (×200 magnification) of reseeded organoids in untreated, IFN-γ-treated, and IFN-γ + HGF-treated conditions are shown in the lower panel. The treatment scheme is depicted in the upper-left panel, and the numbers of organoids regrowing are quantified in the upper-right panel. FM full medium. *n* = 5 independent organoid lines. I IFN-γ-treated organoids. *n* = 5 independent organoid lines. I + H: IFN-γ + HGF-treated organoids. *n* = 6 independent organoid lines D differentiated organoids. *n* = 8 independent organoid lines. H HGF-treated organoids. *n* = 8 independent organoid lines. Scale bar: 100 μm. **b** Live imaging (×200 magnification) showing growth of progenitor cells in untreated, IFN-γ-treated, and IFN-γ + HGF-treated conditions. Scale bar: 30 μm. **c** Actin/phalloidin staining (×600 magnification) demonstrates cell polarity in progenitor cells in untreated, IFN-γ-treated, and IFN-γ + HGF-treated conditions. *n* = 6 independent organoid lines. Scale bar: 30 μm. Assessment of proliferative activity in progenitor cells in untreated, IFN-γ-treated,

and IFN-γ + HGF treated conditions by immunofluorescence (×600 magnification) of EdU labeled S-phase cells (*n* = 10 independent organoid lines) (**d**), of Ki67 positive proliferative cells (*n* = 7 independent organoid lines) (**e**), and of the cell cycle arrest marker p21 (*n* = 8 independent organoid lines) (**f**). Scale bar: 100 μm. **g** Co-staining of KI67 and KRT20 (×600 magnification) in untreated, IFN-γ-treated, and IFN-γ + HGF-treated organoids. Scale bar: 20 μm. *n* = 7 independent organoid lines. All quantifications are presented alongside the respective images. **h**, **i** Differentiated organoids with or without IFN-γ treatment or with IFN-γ + HGF treatment were counted (**h**), and the numbers of organoids regrowing were counted after reseeding them into FM (**i**). *n* = 3 independent organoid lines. Diff. differentiated. All *p*-values were determined using one-way ANOVA (two-sided) with no adjustments for multiple comparisons. Data are represented as mean ± SD. See also Supplementary Fig. 7. Source data are provided as a Source Data file.

into a new flask between each repetition. Enriched crypts were seeded at a ratio of ten crypts per 1 µl basement membrane matrix (cultrex type 1#: 3433-010-R1). The colon organoids were cultured in Advanced Dulbecco's Modified Eagle's Medium/F12 (#12634, Gibco) containing 10 mM HEPES (#15630056, Gibco), 2 mM GlutaMAX (#35050061, Gibco), 1.25 mM N-acetylcysteine (#A9165, Sigma), 25% R-spondin 1 conditioned medium, 1× B-27 (#17504044, Gibco), 1× N2 (#17502048, Gibco), 500 nM A83-01 (#616454, Merck), and 1% penicillin/streptomycin (#15140122, Gibco), 3 µM CHIR-99021 (#S1263, Selleckchem), 0.328 nM sWnt (#N001, U-Protein Express), 50 ng/ml mEGF (#PMG8043 Invitrogen), and 100 ng/ml mNoggin (#250-38, Pepro-Tech). 10 µM Y-27632 dihydrochloride (# M1817, AbMole) was added after passaging and after the initial seeding. Organoids were expanded by single-cell passaging following incubation in Triple E (#12604013, Gibco) at 37 °C for 5 min. After washing in PBS/0.1% BSA, cells were seeded in basement membrane extract and cultured as described above. Differentiation was induced after 3 days of culture by changing to culture medium without sWnt and CHIR or with mEGF and mNoggin. IFN-γ treatment was carried out with 50 ng/ml mouse IFN-γ (#485-MI, R&D) for the indicated times.

### Live imaging
To perform time-lapse imaging, organoids were grown on a glass-bottom 96-well culture chamber (Ibidi, #89626) and imaged using an Observer 7 microscope (Zeiss) equipped with a scanning table and $CO_2$ and temperature control. For each condition, a z-stack with ten slices at a 90 µm distance was imaged every 20 min. Images/videos were processed using IMARIS.

### Primary murine colon stromal cell culture
Crypts were dissociated from tissue fragments for primary murine colon organoid culture. The fragments were washed four times with PBS and shaken after every wash to get rid of epithelium. Once the supernatant was clear, the tissue fragments were cut into tiny pieces and incubated in calcium and magnesium-free HBSS (Gibco) containing Liberase TL (1 unit/ml; Roche) and DNase I (1 unit/ml; Invitrogen) at 37 °C for 1 h, with pipetting every 10 min. Every 20 min, the digested fraction was collected and put into an ice-cold stromal cell medium containing Advanced DMEM/F12, 10% fetal bovine serum (FBS, Gibco), 100 U/ml penicillin/streptomycin, and 10 µM Y-27632. The colon stromal cells were seeded on 75 cm² flasks and cultured in a stromal cell medium.

### MDCK cell culture
MDCK cells were cultured in DMEM (Gibco), 10% fetal bovine serum (FBS, Gibco), 100 U/ml penicillin/streptomycin, seeded on 8-well glass bottom slides (Ibidi, #80827), and treated with 100 ng/ml hIFN-g (R&D) and the indicated concentration of HGF for 24 h.

### Generation of colon assembloids
Assembloids were generated as previously described[2]. Briefly, 3–4-day-old colon organoids were collected by physically pipetting with organoid harvesting solution (R&D) and put on ice for 30 min. When the Matrigel was depolymerized completely, the intact organoids were gently washed with Advanced DMEM/F-12 and allowed to settle by gravity (or centrifuging at 100 g for 1 min at 4 °C). The supernatant was removed, and the organoids were resuspended in Matrigel. The primary colon stromal cells cultured in 75 cm² flasks were treated with 5 ml TrypLE and incubated for 15 min at 37 °C. The dissociated cells were washed with Advanced DMEM/F-12 and centrifuged at 400×g for 5 min at 4 °C. The cell number was determined, and the cells were resuspended in Matrigel (1.5 × 10⁵ cells per 3.5 µl Matrigel). After being sprayed with 70% (v/v) ethanol, a 2 cm×2 cm sheet of Parafilm was placed in a 100 mm petri dish, with several drops of PBS placed around the Parafilm to maintain humidity. A 2 µl Matrigel droplet containing

stromal cells was placed on the Parafilm using a pipette. Then, 0.5 µl of organoids with 1.5 µl of stromal cells were immediately and slowly added into the droplet (4 µl Matrigel per assembloid in total). Ideally, organoids should be in the center of the droplet, and the number should be comparable. The petri dish containing 6 assembloids was covered with a lid and incubated at 37 °C for 10–15 min to solidify the gel. The assembloids were removed from the Parafilm using fine forceps and put into a 12-well plate (6–8 assembloids per well). Unless otherwise indicated, each well contained 1.5 ml organoid medium without sWnt, CHIR-99021, R-spondin 1, or noggin. The plate was put onto an orbital shaker (Edmund Buehler) at 115 rpm in the cell incubator. After 4 days of shaking culture, the assembloids were collected for further analyses.

To visualize Axin2+ and KRT20+ cells and their immediate progeny, the assembloids were generated using epithelial cells from *Axin2CreERT2/Rosa26-tdTomato* and *Krt20CreErt2/Rosa26-tdTomato* mice and treated with 1.5 µM 4OH-tamoxifen for 30 h.

### EdU labeling
EdU labeling was performed using the Click-iT™ EdU Cell Proliferation Kit for imaging (Thermo Fisher Scientific, C10337). Organoids or assembloids were treated with 10 µM EdU for 2 h, followed by fixation with 4% paraformaldehyde and paraffin embedding. Sections were deparaffinized and rehydrated, followed by an antigen retrieval step. After permeabilization with 0.5% Triton X-100 for 20 min, sections were subjected to the Click reaction using Alexa Fluor™ 488 azide for 30 min, followed by primary antibodies against rabbit anti-KRT20 (Cell Signaling Technology, 13063, 1:100), mouse anti-E-cadherin (BD, 610181, 1:200), goat anti-RFP (Rockland, 200-101-379S, 1:200) overnight, followed by a washing step, incubation with secondary antibody (Alexa 488, Cy3, or Alexa 647; Jackson Immunoresearch, 1:300) and counterstained with DAPI (Sigma, 1:300) for 2 h at room temperature. Images were acquired using an Observer 7 microscope (Zeiss) and LSM980 (Zeiss). Images were analyzed with ZEN 3.4 (Zeiss) and ImageJ 2.0.0 (Fiji).

### RNA isolation and RT-PCR
RNA was isolated using the NucleoSpin RNA isolation kit (Macherey & Nagel, 740955) according to the manufacturer's instructions. cDNA was generated using the iScript cDNA synthesis kit (Bio-Rad, 1708891), and qPCR was performed with SYBR-green (Thermo Fisher, A25741) and the QuantStudio Real-Time PCR System (Thermo Fisher). The data were collected with QuantStudio 3 Real-Time PCR Software v1.7.1. Fold-change expression was determined following the deltaCC method and normalized to β-*actin*.

The following mouse primers were used: β-*actin* forward 5′-GCCACTGTCGAGTCGCGT-3′, reverse 5′- GATACCTCTCTTGCTCTGGGC-3′; *Bmp2* forward 5′- GACTGCGGTCTCCTAAAGGTCG-3′, reverse 5′-CTGGGGAAGCAGCAACACTA-3′; *p21* forward 5′- TGCCGAAGTCAGTTCCTTGTG-3′, reverse 5′- GCCATTAGCGCATCACAGTC-3′; *Lgr5* forward 5′-CCTACTCGAAGACTTACCCAGT-3′, reverse 5′-GCATTGGGGTGAATGATAGCA-3′; *Krt20* forward 5′-GTCCCACCTCAGCATGAAAGA-3′, reverse 5′-TCTGGCGTTCTGTGTCACTC-3′; *Xaf1* forward 5′- GAAGCTTGACCATGGAGGCT-3′, reverse 5′- GTGCTGTTGGCTTTCCTTGG-3′; *Axin2* forward 5′- TGACTCTCCTTCCAGATCCCA-3′, reverse 5′- TGCCCACACTAGGCTGACA -3′; *Mki67* forward 5′-CCATCATTGACCGCTTCCTTTAG-3′, reverse 5′-TCTTTGAGCCATCTGAGGCA-3′; *Hgf* forward 5′-TGATTCTTTCAGCCCGGCAT'-3′, reverse 5′-TTCAGTAATGGGTCTTCCTTGGT-3′; *Cxcl11* forward 5′-GGAAGGTCACAGCCATAGCC-3′, reverse 5′-GTCCAGGCACCTTTGTCGTT-3′; *Ifit3* forward 5′-CTGAACTGCTCAGCCCACAC-3′, reverse 5′-GGCCTTGCTTTCGCCATCTA-3′, *Muc2* forward 5′-CAAACCTGTGCGTGTTCCTG-3′, reverse 5′-TCTCGTGGCGCACAATAAGT-3′; *Math1* forward 5′-GTTGCGCTCACTCACAAATAAGCG-3′, reverse 5′-TGGCATTGAGTTTCTTCAAGGCG-3′; *Itf* forward 5′-TGGGATAGCTGCAGATTACGTTGG-3′, reverse 5′-TTTGAAGCACCAGGGCACATTTGG-3′.

## Immunofluorescence

Paraffin-embedded tissue or assembloids sections were deparaffinized, rehydrated and, following an antigen retrieval step, incubated with primary antibodies against rabbit anti-KRT20 (Cell Signaling Technology, 13063, colone D9Z1Z, 1:100), rabbit anti-Kl67 (Cell Signaling Technology, 9129S, clone D3B5, 1:300), rabbit anti-non-p-YAP1 (Abcam, ab205270, clone EPR19812, 1:200), rabbit anti-p21 (Abcam, ab188224, clone EPR18021, 1:300), rabbit anti-YAP (Cell Signaling Technology, 14074, clone D8H1X, 1:200), rabbit anti-cleaved caspase3 (Cell Signaling Technology, 9661, clone Asp175, 1:200), mouse anti-E-cadherin (BD, 610181, clone 36/E-Cadherin, 1:200), rabbit anti-chromogranin A (Abcam, ab283265, RM1025, 1:300), Alexa Fluor 647 fluorophore conjugated phalloidin (Life Technologies, A22287, 1:100) or goat anti-RFP (Rockland, 200-101-379S, clone 234aa, 1:200) overnight, followed by a washing step, incubation with secondary antibody (Alexa 488, 715-546-150; Cy3, 711-166-152; or Alexa 647, 705-605-003) (Jackson Immunoresearch, 1:300) and counterstained with DAPI (Roche, 1:300) for 2 h at room temperature. Samples were imaged with the confocal laser scanning microscope TCS SP8 (Leica) and Observer 7 microscope (Zeiss), and LSM980 (Zeiss). Images were analyzed with Leica Application Suite X (LAS X, Leica), ZEN 3.4 (Zeiss), and ImageJ 2.0.0 (Fiji), and IMARIS 9.8.

## TUNEL staining

Click-iT™ Plus TUNEL Assay Kits (C10617, Thermo Fisher) were used for TUNEL staining. According to the instructions, sections were incubated in a 60 °C heating plate for 1 h and dewaxed in 2× Roti for 5 min each, followed by 100 % ethanol, 96 % ethanol, 85 % ethanol, 70 % ethanol, 50% ethanol for 4 min each, 0.85% NaCl for 5 min, and PBS for 5 min. The slides were re-fixed with 4% paraformaldehyde for 15 min at 37 °C and washed twice in PBS for 5 min each, followed by the permeabilization reagent (proteinase K) incubation for 15 min at room temperature. The slides were washed in PBS for 5 min, and the re-fixation was repeated with 4% paraformaldehyde for 15 min at 37 °C. After the re-fixation, the slides were placed in deionized water. TdT reaction buffer was added to the sections and incubated at 37 °C for 10 min, followed by TdT reaction mixture (TdT reaction buffer, EdUTP, and TdT enzyme) incubation at 37 °C for 60 min. The slides were cleaned with deionized water and incubated with 3% BSA and Triton-X100 in PBS for 5 min. After washing once in PBS, the slides were applied with Click-iT Plus TUNEL reaction cocktail (Click-iT Plus TUNEL Supermix containing Click-iT Plus TUNEL reaction buffer, Copper protectant, and Alexa Fluor picolyl azide; Click-iT Plus TUNEL reaction buffer additive) for 30 min at 37 °C in the dark. The slides were cleaned with 3% BSA PBS for 5 min, then incubated with DAPI (Roche, 1:300) for DNA staining. After staining and washing, the samples were mounted with VectaMount aqueous (Vector) on slides with cover glasses (Marienfeld). Imaging was carried out by the Zeiss Observer 7 microscope, Zeiss LSM980 confocal microscope, and Leica TCS SP8 confocal microscope. The EVOS M5000 imaging system was used for brightfield images. All images were evaluated by Leica Application Suite X (LAS X), ZEN 3.4, and ImageJ 2.0.0 (Fiji).

## Multiplex immunofluorescence

Sequential immunostaining was performed on 4-μm-thick formalin-fixed paraffin-embedded colon tissues or assembloid sections. Deparaffinization, rehydration, and antigen retrieval steps were performed as for immunofluorescence labeling. Antibody elution was performed between each staining cycle using a 2-mercaptoethanol/SDS buffer. The following primary and secondary antibodies and dilutions were used: rabbit anti-KRT20 (Cell Signaling Technology, 13063, colone D9Z1Z, 1:100), rabbit anti-non-p-YAP1 (Abcam, ab205270, clone EPR19812, 1:200), rabbit anti-p21 (Abcam, ab188224, clone EPR18021, 1:300), rabbit anti-Kl67 (Cell Signaling Technology, 9129S, clone D3B5, 1:300) and secondary antibody (Alexa 488, 715-546-150; Cy3, 711-166-152; or Alexa 647, 705-605-003) (Jackson Immunoresearch, 1:300). Nuclei were counterstained with DAPI (Sigma, 1:300) for 2 h at room temperature. Samples were imaged with the Observer 7 microscope (Zeiss) and LSM980 (Zeiss). Images were analyzed with ZEN 3.4 (Zeiss).

## Single-molecule RNA ISH

For single-molecule ISH, 5 μm-thick tissue or assembloid sections were used. RNA in situ detection was performed according to the manufacturer's protocol with the RNAscope Red Detection Kit (Advanced Cell Diagnostics, 322360). The following probe target regions were used for detection: for *Bmp2* 854–2060, for *Hgf* 1120–2030, for *Lgr5* 2165–3082, for *Anxa1* 187–1148, for *Axin2* 330–1287, for *Ppib* (positive control) 98–856. Quantification analysis was performed using ImageJ software with image deconvolution for Fast Red signal selection, and was automatically calculated. To only include the signal within epithelial regions, epithelial regions were circled, and the signal within the area was calculated automatically.

## Single-cell RNA-Seq and analysis

Organoids were grown and treated as outlined above and incubated in Triple E (#12604013, Gibco) at 37 °C for 5 min. After washing in PBS/0.1% BSA, single organoid cells were passed through a 40 μm cell strainer and centrifuged at 400×*g* for 5 min at 4 °C, re-suspended in PBS/0.1% BSA, and supplemented with DNase I (1 unit/ml) followed by FACS to deselect dead cells using DAPI. Each sample was run with Chromium single-cell kits (10× Genomics), following the manufacturer's instructions.

Reads from scRNA-Seq of colon epithelial organoids with and without IFN-γ treatment were mapped against the mouse genome version mm10 using CellRanger v6.0.1. Due to high background, cells with less than 5000 UMI counts, less than 400 detected features, or more than 15% mitochondrial reads were discarded. Samples were normalized in the R package Seurat using standard settings and integrated using CCA. Neighborhoods and UMAP embedding were computed using the first 20 principal components on the top 2000 variable genes and clusters defined using a resolution setting of 0.5. Cluster identities were assigned based on known epithelial subpopulation markers. Finally, similar clusters were joined to obtain an easily interpretable classification.

Published scRNA-Seq data of mouse stroma in healthy controls or after DSS treatment were obtained from GSE114374[28]. Cells with less than 5% mitochondrial reads and between 500 and 5000 detected features were normalized in Seurat and preprocessed and integrated as above, with a resolution setting of 0.1. Cluster identities were assigned based on typical marker gene expression. Stromal and myofibroblast clusters were extracted and reclustered on the first ten principal components with a resolution of 0.15. Stromal clusters were labeled based on marker gene expression from ref. [28] and ref. [66]. scRNA-Seq data of mouse epithelial cells in healthy controls or after DSS were obtained from SRA[25] and processed with CellRanger v6.0.1. Cells with > 40% mitochondrial reads, less than 500 or more than 6,500 detected genes, and less than 100 or more than 60,000 UMI counts were excluded, similar to parameters used in the original publication. We first identified and filtered for epithelial cells and then integrated all samples from uninjured and mock-treated DSS conditions using CCA. Clusters were defined using the first 15 principal components and a resolution of 0.3. Cluster identities were assigned based on typical marker gene expression.

The transcriptional response to IFN-γ was assessed using genes from MsigDB (version MSigDB 2023.1[67], gene set HALLMARK_INTERFERON_GAMMA_RESPONSE by R package GSVA[68]. Sequencing data from this study are deposited at GEO (GSE255366). For BMP response, we used the geneset SUH_COEXPRESSED_WITH_ID1_AND_ID2_UP.

## Expression and purification of two-chain HGF/SF (tc-HGF/SF)

A full-length human HGF/SF was cloned into the expression vector (pA71, carrying human cytomegalovirus promoter sequences and a hygromycin-resistant gene) and transfected into the mouse myeloma line NS0 for expression of fully glycosylated tc-HGF/SF or into the mutant Chinese hamster ovary cell line Lec3.2.8.1 (EGT 92/A50.373) for expression of minimally glycosylated tc-HGF/SF. Stable integrants were selected in the presence of 1 mg/ml hygromycin and screened for expression of HGF/SF using a biological assay[37]. Supernatants of large-scale (10–20 l) cultures of NS0 expanded in DMEM (NS0) in the presence of 1% FCS were cleared by centrifugation and dialyzed/concentrated to a final volume of ~500 ml of 0.025 M phosphate, 0.15 M NaCl, pH 7.4 using a bench-top cross-flow instrument (Flowtech Inc.) equipped with a 10 kDa cut-off Sartocon Hydrosart dialysis cassette (Sartorius). The tcHGF/SF CHO lec3.2.8.1 production cell line (EGT 92/A50.373) was cultivated in a 2.5 l stirred tank bioreactor in perfusion mode (final perfusion volume 20–40 l) in ProCHO5 medium supplemented with 7.5 mM Gln. The harvested culture supernatant was diafiltrated against PBS pH 7.4 buffer and concentrated using a Prostak ultra-filtration unit equipped with Millipore Pellicon 2 Cassettes (10 K PLCGC-C 0.5 SQ.M). The final concentrate (2 l) was filtered through Sartolab-P20 filters (0.2 μM). The diafiltrated and concentrated supernatants were loaded onto a Heparin-Sepharose Fast Flow column (GE Healthcare). Bound proteins were eluted with a gradient of NaCl (0.15–2.0 M) in 0.025 M phosphate pH 7.4 and fractions containing tc-HGF/SF—as confirmed by the biological assay[37] and SDS-PAGE—were dialyzed against 0.05 M MES, 0.25 M NaCl pH 6.0 and applied directly to a Mono S HR column (GE Healthcare). tc-HGF/SF was eluted with a gradient of NaCl (0.25–2.0 M)[69]. Purity of tc-HGF/SF after cation exchange chromatography was consistently higher than 95% by SDS-PAGE. The concentrated supernatant of culture of the CHO lec3.2.8.1 cell line (EGT92/A50.373) was loaded on a Heparin-Sepharose FF column (GE Healthcare, XK50 with a bed volume of 25 ml). The column was equilibrated with 50 mM Tris-HCl, pH 8.0, 150 mM NaCl. After loading the sample overnight, the column was washed in two steps at 400 mM NaCl, 50 mM Tris-HCl pH 8.0 (10 CV) and 600 mM NaCl, 50 mM Tris-HCl pH 8.0 (10 CV). The tc-HGF/SF was eluted at 1 M NaCl: 50 mM Tris pH 8.0. The tc-HGF/SF fractions were pooled and dialyzed overnight against 5 l (50 mM Tris-HCl pH 8.0; 250 mM NaCl). To remove residual sc-HGF/SF, the protein was completely processed to tc-HGF/SF by the addition of 1/40 HGFA (HGF-activator) in a molar ratio (HGFA was produced and purified from baculovirus-infected insect cells). To remove HGFA, the sample was purified after incubation and dialysis by chromatography on a 5 ml HiTrap Heparin column (GE Healthcare) using the same step gradient as described before. The pooled tc-HGF/SF fraction was dialyzed against 50 mM HEPES pH7.4; 250 mM NaCl and loaded on a 1 ml MonoS GL column (GE Healthcare) equilibrated with 50 mM HEPES pH7.4; 250 mM NaCl. The bound protein was eluted using a linear gradient from 250 mM NaCl to 1 M NaCl in 50 mM HEPES buffer pH7.4), pooled and dialyzed against 50 mM HEPES pH 7.4; 600 mM NaCl (Supplementary Fig. 8a). The fractions were concentrated to 2.0 mg/ml using Vivaspin concentrators with a MWCO of 10 kDa. The concentration of tcHGF/SF was calculated using Abs280, 0.1% = 1.85. Samples were split into 40 μl aliquots, shock frozen in liquid nitrogen, and stored at −80 °C.

## Reporting summary

Further information on research design is available in the Nature Portfolio Reporting Summary linked to this article.

## Data availability

All sc-RNAseq data generated in this study have been deposited in the National Center for Biotechnology, Gene Expression Omnibus (GEO) under accession code GSE255366. The previously published sc-RNAseq data are available in the GEO under accession codes GSE201723 and GSE114374. All other data supporting the results presented herein are available within the article and Supplementary Information. Source data are provided with this paper.

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

## Acknowledgements

The authors thank Janine Wolff, Yvonne Giesecke, and Patrick Seidler for excellent technical support; we thank Caroline Braeuning and members of the Scientific Genomics Platform of the BIMSB/BIH for technical support; we gratefully acknowledge Felix Heymann from the Department of Hepatology and Gastroenterology of the Charité for imaging support. We thank Anja Kühl and members of the Charité Core Facility for Histopathology for sample processing. We are grateful to Rike Zietlow for editing the manuscript. We thank Frank Tacke and the Tacke lab for the infrastructural and scientific support. The work was supported by the DFG Emmy Noether grant to M.S. (Si-1983/4-1), the ERC starting grant (REVERT agreement number 101040453) to M.S., the Einstein Foundation grant within the EC3R initiative to M.S., and BMBF grant 01EJ2206A (PACETherapy) to M.S. L.L. was awarded a fellowship (202008350143) from the China Scholarship Council.

## Author contributions

The project was conceived and designed by J.H., L.L., and M.S. L.L. and J.H. performed most of the experiments under the supervision of M.S. J.H. and L.L. performed immunostaining and imaging. H.B., J.H., and L.L. performed scRNA-seq and subsequent analyses. L.L. and M.L. processed assembloids for histological experiments. L.L., J.H., and S.M. performed qPCR. H.J., J. v d H., and E.G. provided support for HGF experiments. S.B., G.B., and M.L. provided imaging support. M.S. acquired funding and provided infrastructure and guidance for this project. The manuscript was written by J.H., L.L., and M.S.

## Funding

## Competing interests

The authors declare no competing interests.
