## [Transparent Peer Review file · Nature Communications]

Extrusion of BMP2+ surface colonocytes promotes stromal remodeling and tissue regeneration

Corresponding Author: Professor Michael Sigal

Version 0:

Reviewer comments:

Reviewer #1

(Remarks to the Author)
General comments

In the submitted manuscript, Heuberger and colleagues examine responses of the colonic epithelium to damage as modelled in mice using an inflammatory challenge (DSS) or via addition IFN-gamma to primary cultures, to model the release of this cytokine by lymphocytes in response to damage to the epithelia.

The process of intestinal epithelial damage and regeneration from multiple fairly plastic cellular compartments has been well documented previously – although detailed functional validation of molecular mechanisms regulating this process are still warranted. This study focusses on the response of the epithelia to IFN-gamma and the role of BMP2-expressing colonocytes during repair. It builds on the team's prior expertise and makes use of their recently developed assembloid system to model mature colonocyte behaviour in the presence of co-cultured stromal cells in vitro, together with more conventional organoid and mouse models. The imaging and staining is generally of a high quality.

The important role of BMP/TGF β signalling in intestinal epithelial health and the regenerative process have been previously well established, but the focus has primarily been on signalling at the crypt bottom around the intestinal stem cell niche and the BMP gradient up the crypt (recently reviewed for eg in PMID: 35611490). However the idea of a decision point for differentiation of the normal intestinal epithelia that is regulated by BMP signalling and anti-microbial responses is well elucidated and relevant here (PMID: 35235783). Equally the key function of YAP signalling in regeneration and previously identified upregulation by IFN-gamma (eg. PMID: 33606996) are known.

Overall, while the findings are of interest, the manuscript would benefit from consolidation of repetitive sections and improved clarity of the novel findings for the reader. Claims in this manuscript were also somewhat inflated and not consistent with the data provided in the manuscript, but understandably drawing on what is known in the field. The novelty of this work then appeared fairly incremental to this reviewer, and unfortunately not at a level commensurate with publication in Nature Communications without major review.

Major points:

1. The concluding statement of many results sections and section headings/figure titles/paper title are inflated and not appropriate as written.

a. Eg. Figure 1 legend title - Line 95 – ‘...IFN-gamma signaling is an endogenous driver of stem cell loss...’ The results show decreased expression of stem cell markers that does not occur in the KO mice upon DSS treatment, not analysis of stem cell behaviour or lineage/label retaining cell tracing, or even multiple time points. Unclear how ‘stem cell loss’ is demonstrated when regenerative progenitors can lose Lgr5.

b. Line 127/9 – ‘number of apoptotic cells was rather low, likely because they are quickly extruded...rapid loss of differentiated colonocytes’ In the absence of multiple time points after DSS treatment, and co-staining with an epithelial marker, it is difficult to assess the true level of apoptosis across time in the epithelium. Unclear on what basis the rapidity, or even evidence of cell loss (as opposed to cell arrest as p21 is later shown to be upregulated with DSS treatment) assumption, is being made, based on the data provided in this section. Also line 344/352 discussion – process is claimed to be accelerated/massive exfoliation – based on what data from this paper or other citations? Line 354-361 is supposition and not in keeping with the data presented in the paper.

c. Paper title & Fig 2 title – ‘Fig. 2: IFN- γ -induced removal of the differentiated compartment and transformation of progenitors into pre-regenerative Krt20+ colonocytes.’ Title needs to reflect what is shown in paper and figure. How is removal of KRT20

compartment shown exactly & how does it fit with Krt20+ cells being increased in vivo in Fig 1? How are progenitors shown to be changed into Krt20 colonocytes? Paper title – how is ‘programmed extrusion’ shown? Other points - Fig 2j/k – why is lineage tracing with an Lgr5-cre line not included? The population represented by Axin2-lineage is not described. Quantification is unclear-Krt20 cells as a % of what? DAPI+ cells? Terminology of a 50% increase in Axin2-traced cells is confusing, given the traced cells are presented as a %-consider altering wording. Fig 2k&l green marker is lineage cells, not KRT20/AXIN2 protein+. Label should make this distinction clear for the reader. Top of IFN-gamma treated image is without cells-how was region selected? This does not match other images of IFN-gamma treated assembloids in Fig 2(?) It appears as though the Axin2-lineage cells are likely the predominant epithelial population following IFN-gamma treated but this is not detailed in the summary for this section, only the KRT20 cells? How does this result (and the reduced KRT20 IF in 2h +IFN-gamma) fit with the in vivo increase in KRT20+ population following DSS treatment? What does box plot denote for this? 25th/75th percentile? Is n=5 from biological replicates or fields of view from same experiment? Line 191/199 – stated that the organoid and assembloid data ‘revealed the de novo production of colonocytes’ How? Ki67 goes down, so lineage alteration (or simply colonocytes that don’t express Bmp2) rather than ‘de novo production’?? Need to be more careful with terminology. As written it is confusing for the reader.

d. Fig 5 does not functionally remove BMP2 signalling and yet makes claims about ‘loss of BMP2’ or ‘BMP signalling’ (also discussion line 301, 337, 341 etc). Overstatement currently, but fairly straight-forward to provide more data on this point. Line 255/264 –‘loss of BMP2’ can be more effectively modelled using Bmp receptor knockout lines or addition of BMP antagonists to support this claim, rather than exogenous additional of Bmp2 ligand. Signaling can be examined by Smad deletion etc. Lack of addition of ligand does not always equate to situations when signaling is ‘lost’.

e. Evidence for ‘extrusion’/death of colonocytes? Fig 3c – where is proof that cells are dead but Fig 3d cells are not, given caspase-3 stain in other sections of paper & claims that cells ‘die’ vs live? Line 299 ‘inducing apoptosis of BMP-2 expressing colonocytes’-what is the evidence presented for this? Soften wording or provide data to support.

f. Line 293 – ‘regeneration that is independent of Lgr5+ stem cells’ and Line 304 in discussion. Yes, this is likely given the body of literature on this subject, however, the data presented do not present information about dependency on the Lgr5 stem cell population, only Lgr5 marker levels. To make this claim would require lineage tracing or similar technique (Lgr5 ablation) to examine contribution from Lgr5 cells. Re-word or provide new data.

g. Discussion Line 313 ‘orchestrated endogenous process that extends to undamaged tissue regions...’ where are ‘undamaged’ tissue regions included in the data? Overstatement? Ext data 4d is not referred to in the text and contains an ‘ulcerated’ region of colitis. Please provide high resolution H&E or other image to show region of ulcer and describe results and conclusions in text or remove.

h. Discussion line 322 – ‘KRT20+ cells... can be reprogrammed to proliferate and enter the regenerative state’ Over-claim, please re-word to show assumptions/suggests etc. The data do not show this – there is one lineage tracing experiment that does not show KRT20+ precursor give rise to proliferative cells.

i. Line 324 ‘...do not express BMP-2, are unresponsive to BMP signaling’ There remains some (ie not none) Bmp2 expression in Fig 2a colitis in WT and organoids +IFN-gamma 2e-temper wording.

2. At the molecular level, can additional information be provided on how HGF is upregulated in response to IFN-gamma treatment? How does HGF then result in release of epithelial cells from arrest?

3. Discussion should include previous literature on the role of BMP4/BMP antagonists in regeneration (eg PMID: 33819486)

Minor points

1. scRNAseq analyses. Line 143/4– Fig 2b has no statistical evaluation of differential expression of Krt20 & Bmp2 included for the reader- depict as violin plots or similar with appropriate statistical evaluation. Line 149/50 - Fig 2e – no statistically sig difference in Krt20 or Bmp2 is shown after IFN treatment, re-word. Line 151/2 – ‘increased expression of transit-amplifying cell markers’ claimed, yet again no stats, two markers in this group appear to decrease (including Ki67 which fits with subsequent staining image). Line 195 ‘differentiation markers were increased’-needs stats. Fig 5h, line 263/4 needs correlation analysis.

2. Fig 2d-intensity of DAPI and E-cad staining is not consistent across all panels

3. Line 252- need to explain that the ‘IFN-gamma response’ is based on transcriptomic measurement using a scoring metric.

4. Fig 3a, d n=? Fig 3e – n=3-is that independent expts with same colon organoid line? Error bars denote?

5. Fig 4 legend title – Induction of [partial? ie not proliferation?] regenerative features ..

6. Fig 5a n=30 – organoids? Independent replicates or all in one expt? What do dashed lines denote in violin plots? Fig 5e n=3 – mice? Error bars denote?

7. Ext data fig 2a n=3 means 3 independent cultures of the same organoid line? 2e-legend but IF of p21 in figure is missing and not mentioned in text.

8. Ext data fig 4 – IL22 not expressed or below QC cutoffs?

9. Line 324 typo ‘homoeostatic’

Reviewer #2

(Remarks to the Author)

Heuberger et al. investigate the role of IFN- γ -dependent inflammatory response in colonic epithelial injury. The authors induced epithelial damage by exposing IFN γ R knockout and wild type mice to DSS to study the effects on the present cell types and marker expression. In an acute phase of injury, the authors find a surprising IFN γ R-dependent downregulation of proliferation and stem cell markers and an increase of KRT20+ colonocytes. Experiments in colon organoids and assembloids containing both colonic epithelial and stromal cells revealed an induction of apoptosis and reduction of proliferation upon IFN γ treatment. Using scRNA seq, IHC, IF, and qPCR the authors find a reduction in stem and secretory cell identity and an increase of a KRT20+ colonocyte type with reduced BMP-2 expression. Using a lineage tracing approach, the authors argue that IFN γ treatment promotes the loss of differentiated cells and the de novo generation of these KRT20+, BMP-2-low colonocyte that are in cell cycle arrest ("pre-regenerative colonocytes"). According to the authors' model, the prevalence of this transient cell type causes a re-organization of the mesenchymal niche stimulating the latter to express HGF, which in turn releases pre-regenerative colonocytes from cell cycle arrest and induce proliferation to promote tissue regeneration.

The findings reveal a complex interaction between epithelial and mesenchymal cells in response to injury-caused inflammation and a surprising transient pre-regenerative state at an initial injury phase. The study is well-conducted and follows a logical experimental design supporting the proposed model. The results of this study are novel and will be of interest for a broad audience and suitable for publication in Nature Comm.

There are a few points the authors should address to further validate the proposed model:

The scRNA sequencing was done on organoids in expansion medium. The authors state that in expansion medium, differentiated cells are not present. Yet, they analyze different (mature?) cell types in this medium. Why was this analysis not performed e.g. in assembloid samples that contain the relevant cell types (stem, TA cells, colonocytes, etc.)? It is unclear how well the data characterizes the behavior of mature cell types (as suggested by the applied labels) or rather only apply to progenitor cells. Clarification from the authors would be appreciated.

Additionally, the scRNA seq seems to show that all cell types and multiple genes are affected by IFN γ treatment. It would be good if the authors could clarify their selection criteria/rational for their further focused investigation e.g. on BMP-2. The UMAP of both, control and IFN γ -treated organoid sample, should be shown. Does the colonocyte population shift/separate more apart or closer relative to the progenitor population?

In general, the manuscript would benefit from a deeper characterization of the identified transient KRT20+, BMP-2-low colonocyte. How do they compare to mature colonocyte and TA cells, respectively? I would suggest profiling these cells at the transcriptomic level using scRNAseq ideally from tissue, which should be well possible given their high abundance upon DSS treatment, or alternatively from assembloids upon IFN γ treatment. Are there any other specific markers that characterize them, and do they express other genes that would support their proposed role in reprogramming the mesenchyme?

The authors suggest that upon IFN γ treatment differentiated cells are lost and colonocytes form de novo from stem cells that are locked in cell-cycle arrest until upon HGF stimulation the tissue enters the regeneration phase and cells proliferate. The authors seem to suggest that it is specifically this population of colonocytes that de-differentiates and regenerates injured tissue rather than remaining stem or TA cells (that might be reduced in number but sustain upon IFN γ treatment or injury-induced inflammation). To validate this hypothesis, the authors should show more directly that it is the KRT20+ (BMP-low) colonocytes that proliferate upon HGF exposure. A co-staining of Ki67 and KRT20 would be essential.

The authors should design an experiment to test the hypothesis at the functional level. They could make use of their lineage tracing models to test if de novo formed KRT20+ colonocytes replicate to form new tissue after HGF treatment or rather the remaining SCs. Can these 'pre-regenerative colonocytes' grow entire organoids from single cells upon HGF treatment?

Do colonocytes dedifferentiate into S/TA cells and lose KRT20 upon HGF treatment? A KRT20 staining upon HGF treatment would be great.

Reviewer #3

(Remarks to the Author)

Dextran sodium sulphate (DSS) treatment is a popular model for colonic epithelial injury. The authors of this manuscript previously reported a role for IFN γ R in a gastric infection model (Kapalczyńska et al, 2022) and here they extend the theme to DSS-injured colon, though much of the study hinges on IFN γ -treated colonic organoids. They propose a model where IFN γ simultaneously promotes loss of BMP2-expressing colonocytes, cell cycle arrest, reduced stem cells, and increased differentiation marker Krt20. These cells are proposed to be "pre-regenerative" (neither stem-like nor differentiated). Upon loss of BMP2-expressing colonocytes, the underlying mesenchyme also decreases BMP and increases HGF signaling. HGF overrides IFN γ -induced effects on organoids, promoting return to epithelial homeostasis following injury.

Major comments:

1. The authors go back and forth between DSS colitis in vivo and IFN γ -treated organoids. Even if there are some parallels, DSS colitis is clearly different from an IFN γ response, and many in vitro and in vivo findings don't align, e.g., secretory (goblet) cells are well preserved in colitis (Ext. Fig. 1a-b) but significantly compromised in IFN γ -treated organoids (Fig. 2b). A conflation of experimental models leads to a convoluted model of unusual phenomena (death and production of selected cell types) that require more rigorous evaluation before being ready for publication.

2. The authors suggest that IFN γ signaling causes rapid colonocyte loss after DSS treatment. The claim relies on a finding that Bmp2 expression is reduced in acute DSS colitis in WT but not in IFN γ R KO mice (Fig. 3a-b, not quantified and not entirely convincing). The authors do not distinguish between expulsion of Bmp2-expressing cells (their interpretation) and the simpler explanation of reduced Bmp2 expression. Experiments in Fig. 3c-d do not distinguish between these possibilities. There are few apoptotic cells and the manuscript presents no evidence for "programmed extrusion of BMP2+ surface colonocytes" (title, etc.). In fact, Krt20+ cells are increased, not decreased, in acute colitis (Fig. 1e). Additional and better studies are needed to support the conclusion that "BMP-2 expressing homeostatic colonocytes die in response to IFN- γ , while IFN- γ induced de-novo colonocytes do not".

3. A key element of the model is that IFN γ -treated colonocytes represent a "new" state, with simultaneous features of proliferation and cell cycle arrest (e.g., cell cycle marker Ccnd1 is increased but MKi 67 is decreased). These and other interpretations are needlessly far-fetched. A simpler explanation is that scRNA-seq data are technically limited (Fig. 2b – e.g., cell proportions from untreated and IFN γ conditions are not indicated) and interpreted too liberally.

4. The conclusion that IFN γ induces a "pre-regenerative colonocyte state" is especially convoluted and superficial. It rests on a series of findings (Anxa1 expression, phosphorylated YAP, p21 expression) that are likely unrelated but which the authors assume to be causally related to one another. No alternative interpretation is considered and no experiments are performed to validate the "pre-regenerative state". No evidence is presented for a presumed transition from "IFN- γ -induced pre-regenerative cells into the proliferative regenerative state".

5. The connection with HGF is plausible and is supported by published data, but alternative possibilities are not considered and the model is assessed superficially. Much is made of the observation that HGF rescues the IFN- γ induced reduction of reseeding capacity when colonoids are grown in full medium but not when they are grown in differentiation medium. There are many possible reasons for that phenomenological difference but the authors consider only one. Their model is not necessarily wrong but easily could be. The necessary level of rigour (entertaining and excluding various possible explanations) is missing from the study.

6. Do the authors propose that IFN γ decreases mesenchymal BMP2 expression? It is difficult from the data to connect epithelial BMP2 loss to the loss of BMP response/increased HGF in the mesenchyme. Could IFN γ itself not be doing both? What is the timeline of HGF expression coinciding with BMP2/Id loss in the epithelium and mesenchyme during acute DSS colitis?

7. Do the authors propose that HGF decreases epithelial IFN γ receptor expression (Figure 6) or a separate pathway that bypasses IFN γ -induced cell cycle arrest?

Technical concerns:

1. Dot counts for Lgr5 and Axin2 (Fig. 1) do not match the images shown. Hardly any Lgr5 dots are visible in most crypts compared to Axin2, yet the graphs show almost similar y-axis values.

2. Colon tissue sections from experimental and control mice are not comparable in many cases (Figs. 1a, 1e, 3a-b, 4a-b). Crypt orientation is vertical in some images, tangential in others. Differences are particularly stark in Fig. 3.

3. Rudimentary phalloidin staining is not adequate to conclude that organoids treated with IFN γ have "more pronounced cell polarity." The finding is soft and may not be relevant.

4. scRNA-seq. Are the authors comparing roughly similar numbers and similar quality from each condition (untreated and IFN γ treatment)? Effects of IFN γ treatment could reflect poor organoid health (e.g., Fig. 2f-g, Ext. Fig. 1d – are apoptotic markers expressed?) rather than specific Bmp2 reduction and other purported changes. The authors use >15% mitochondrial reads as a cutoff (the norm is >5%). Few markers are assessed. One could take any few scRNA markers and interpret the subsequent histological data (Fig. 2d, f, g) with a different lens. The authors should approach these data with stringent QC measures and interrogate meaningful panels of genes rather than single genes (Krt20, Bmp2) that could be misleading in sparse scRNA-seq data.

5. Judging by the dots, Pbx1 seems mistakenly annotated as a progenitor marker (Fig. 2b)

6. Ki67 immunostaining does not look reduced in IFN γ -treated assembloids (Fig. 2g). Krt20 expression looks the same (possibly higher) in untreated assembloids (Fig. 2h). Assembloid data (Fig. 2g-i) are not interpreted objectively. Select and tiny quadrants are shown (other areas show less difference) and findings are not quantified. All immunostains should be quantified.

7. Assembloid data on cell death (Fig. 2i) are not informative (alone) because apoptotic cell burden varies widely across untreated and treated assembloids for a number of reasons. Can the authors clarify and quantify when DSS-exposure

causes apoptosis in vivo? Is the effect abrogated in IFN γ KO mice? Is IFN γ expressed in colonic epithelium, mesenchyme, or both?

Version 1:

Reviewer comments:

Reviewer #1

(Remarks to the Author)

1. Overall the authors have worked to include new data to address concerns and modified the text, however concerns remain about clarity and interpretation of findings. The combination of the different model systems used is still not as clear as the discussion portrays, especially as discrepancies between the models are not addressed in the results or discussion section to guide the reader. At the most basic level, IFN-signalling in colitis in vivo and in vitro IFN treatment of organoids results in increased Krt-20 labelled epithelia. This contrasts to IFN treatment of assembloids reported to decrease/not change percentage of Krt20+ cells. No explanation of why results reported in the different models varies – this is glossed over which doesn't assist reader comprehension. Extrusion of BMP2+ surface colonocytes is still a main focus of the paper but some of the data is unconvincing/conclusions over-stated.

2. Wording over-stated, results images not consistent across manuscript.

a. Line 131 'indicating an increased IFN-g dependent loss of mature colonocytes in acute colitis'

The WT colitis H&E representative image included in Ext data Fig 1g doesn't match the other images included with different stains from this same group in the paper (eg luminal surface Fig 1a/1e-not such obvious 'disintegration'.) How then is this image representative?

This and apoptotic stains are key in vivo data in the paper showing potential INF dependency of the apoptosis/disintegration phenotype. While position of cells is consistent with KRT20+/Bmp2+ colonocytes, suggest remove this claim/re-word unless co-stain of TUNEL or CASP3 with KRT20+/Bmp2 can be provided – presumably 3d DSS timepoint captures this best, rather than d5 (no quantification provided for Ext data fig 1h).

b. Line 176-177 – quantification is now included but is not consistent with the statements in the text – '...IFN-g...reduced numbers of Ki67+...decrease in KRT20+' when quantification now shows Ki67 is increased with IFN-g and KRT20+ not sig different? Is text referring to a specific region not quantified? Unclear how assembloid regions of interest were selected, prone to bias in selection of small region from the large assembloid. Zoomed out versions of the assembloid would be informative for the reader.

c. Fig 3f IFN treated assembloid image does not match any of the other similarly treated assembloid images -authors previously responded that they presumed the shed cells had detached in this image. But now with new images, especially new fig 3h with KRT20 IF, 3j KRT20 tracing in apoptotic cells - why is a region with cells in only half the image chosen for 3f, especially when major conclusion is about 'extrusion' of KRT20+ cells? Are additional timepoints required to capture? No mention of why ACTIN is used for costain in images for the reader (not mentioned in text or figure legend)

d. Ext data video 3 is unconvincing – no quantification or indication of replicate numbers in figure legend. Line 137 'a loss of mature colonocytes' but apoptotic cell death was by the authors admission 'rather low' and KRT20 cells are increased with colitis, so this statement is misleading/one-sided as written.

e. 7 figures not warranted - Fig 2 could be moved to extended data to reduce figure number in manuscript proper

f. Fig 3 - The conclusion from this section does not accurately represent the data, it is very one-sided and overplays the lineage tracing – these are ~2d tracing studies only, so many of the positive cells will be Krt20 or Axin2 expressing rather than progeny. Eg text states 'promoting differentiation of Axin2 progenitors into Krt20+ colonocytes' & 'promotes the generation of new KRT20+ cells' when overall the number of KRT20+ cells is not different with IFN treatment (Fig 3c quantification n.s.), the number of KRT20 lineage traced cells decreases with IFN and doesn't acknowledge the key increase in Axin2 'progenitor' type cells in the culture with IFN treatment.

This flows through to the summary statement at the beginning of Fig 4 text 'The organoid and assembloid data revealed an IFN-dependent differentiation of KRT20+ colonocytes'. The organoid data shows a partial differentiation phenotype. The assembloid data does not support a differentiation increase – apoptosis, crypt reduction, possible increase in Wnt+/Ki67+ population yes but not increased differentiation from data presented.

3. Minor points

a. Line 190 – 'At 24h after induction of KRT20+ colonocyte lineage tracing' – but 3e shows that harvest point was at 54h post-OHT lineage trace induction? Alter wording.

b. Fig 3j – new data – why is this presented split into channels? The rest of the figures are not. Negative control IF for the TdTom lineage stain needs to be shown given false positive problems with dead/dying cells staining as found in centre of assembloid. Good to show no primary or similar with same secondary to confirm validity of stain.

c. Ext data Fig 3a – now included new analysis of a public scRNAseq dataset. Difficult to evaluate. 'Mature colonocyte' and 'immature colonocyte' labels are used, presumably based on original publication, but how were these terms assigned? On what basis? What does that mean for the markers that have been used in this paper (Krt20 in particular) & comparison to 'progenitor/TA'/colonocyte' clusters from Fig 2 scRNAseq? Figure lacks standard information such as p values or other statistic to indicate significance - error bars +/- 2 SEM appear overlapping? How many biological replicates, was total cell number analysed equal, noting these are absolute cell numbers rather than proportion of the total number examined?

d. Line 308- without including analysis of DSS treated Bmpr1aKO mice, this statement has not been proven, yet wording suggests this. Modify wording to 'This suggests that loss of BMP signalling may...'

Reviewer #2

(Remarks to the Author)

I appreciate the experiments the authors conducted to validate their key results, as well as the textual revisions made to temper the boldness of their interpretations. The lineage tracing experiments help supporting the presented model.

I do find the repetitive posting of the very same results in response to the reviewers' comments irritating.

Minor problems with the new experimental data:

Figure 3h: The images and especially the color choice is not ideal. Difficult to appreciate colocalization.

Figure 7g: The quantification shows 40% of Ki67+ cells are Krt20+. In the zoomed image there is not a single Ki67+ cell visible but in the organoid next to the selected one, there are multiple Ki67+ cell visible, and Krt20+ is not visible at all. Did the authors choose a representative image? How heterogenous is the response of individual organoids in the IFN- γ condition, i.e. what percentage of organoids upregulate Krt20?

Version 2:

Reviewer comments:

Reviewer #2

(Remarks to the Author)

The authors have improved their explanations of the functional differences between organoids and assembloids, which should help readers critically evaluate the data. Their explanation for the discrepancy in KRT20+ cell numbers—that de novo KRT20+ cells replace homeostatic cells due to baseline differences between the models—is plausible. However, it remains a weak point that these models reflect in vivo conditions differently for this key finding. A clearer statement acknowledging the limitations of both models would help mitigate criticism. Experimental evidence comparing these systems in this aspect in a targeted way would strengthen the manuscript.

Given the central importance of the loss of KRT20+/BMP2+ colonocytes in acute colitis, apoptosis staining such as TUNEL should be shown as co-staining with key differentiation markers (e.g., KRT20, BMP2), as suggested by Reviewer 1.

The authors have appropriately softened their claims (e.g., regarding BMP signaling and KRT20+ cell extrusion) and revised their conclusions to be more hypothesis-driven rather than definitive, addressing concerns about overstated interpretations.

The justification for using an unrepresentative image (Fig. 3f) in whole-mount imaging remains insufficient. This may leave doubts about data presentation. To prevent inconsistency and improve transparency, zoomed-out views or additional representative images should be provided in the extended data.

Point-by-point response

We would like to thank the reviewers for their constructive comments, which have helped us to improve the manuscript significantly. Below, we have addressed each of the points they raised in turn.

Reviewer #1 (Remarks to the Author):

General comments

In the submitted manuscript, Heuberger and colleagues examine responses of the colonic epithelium to damage as modelled in mice using an inflammatory challenge (DSS) or via addition IFN-gamma to primary cultures, to model the release of this cytokine by lymphocytes in response to damage to the epithelia.

The process of intestinal epithelial damage and regeneration from multiple fairly plastic cellular compartments has been well documented previously – although detailed functional validation of molecular mechanisms regulating this process are still warranted. This study focusses on the response of the epithelia to IFN-gamma and the role of BMP2-expressing colonocytes during repair. It builds on the team's prior expertise and makes use of their recently developed assembloid system to model mature colonocyte behaviour in the presence of co-cultured stromal cells in vitro, together with more conventional organoid and mouse models. The imaging and staining is generally of a high quality.

The important role of BMP/TGF β signalling in intestinal epithelial health and the regenerative process have been previously well established, but the focus has primarily been on signalling at the crypt bottom around the intestinal stem cell niche and the BMP gradient up the crypt (recently reviewed for eg in PMID: 35611490). However the idea of a decision point for differentiation of the normal intestinal epithelia that is regulated by BMP signalling and anti-microbial responses is well elucidated and relevant here (PMID: 35235783). Equally the key function of YAP signalling in regeneration and previously identified upregulation by IFN-gamma (eg. PMID: 33606996) are known.

Overall, while the findings are of interest, the manuscript would benefit from consolidation of repetitive sections and improved clarity of the novel findings for the reader. Claims in this manuscript were also somewhat inflated and not consistent with the data provided in the manuscript, but understandably drawing on what is known in the field. The novelty of this work then appeared fairly incremental to this reviewer, and unfortunately not at a level commensurate with publication in Nature Communications without major review.

We have now consolidated repetitive sections and improved the clarity of what is novel about our findings. While we agree that the signaling pathways that we evaluate here have been studied before individually, our study provides novel conceptual insights into the interplay of these signals involving a cross-talk of epithelial, stromal and immune cells that together govern tissue reprogramming in the context of injury.

Our finding that loss of surface cells is not merely a passive effect of damaging factors, but an active, widespread and most likely adaptive response that relies on communication and a signaling circuitry between the mucosa and the adaptive immune system, as well as the finding that this process impacts the surrounding stroma, are entirely novel.

Major points:

1. The concluding statement of many results sections and section headings/figure titles/paper title are inflated and not appropriate as written.
 - a. Eg. Figure 1 legend title - Line 95 – ‘...IFN-gamma signaling is an endogenous driver of stem cell loss...’ The results show decreased expression of stem cell markers that does not occur in the KO mice upon DSS treatment, not analysis of stem cell behaviour or lineage/label retaining cell tracing, or even multiple time points. Unclear how ‘stem cell loss’ is demonstrated when regenerative progenitors can lose Lgr5.

We have now revised the titles and concluding statements accordingly. The particular statement highlighted by the reviewer in a) has been changed to:

“Thus, we conclude that IFN- γ -signalling suppresses the expression of Wnt and stem cell marker genes Lgr5 and Axin2 in the context of colonic crypt upon injury.” (Line 88-90)

b. Line 127/9 – ‘number of apoptotic cells was rather low, likely because they are quickly extruded...rapid loss of differentiated colonocytes’ In the absence of multiple time points after DSS treatment, and co-staining with an epithelial marker, it is difficult to assess the true level of apoptosis across time in the epithelium. Unclear on what basis the rapidity, or even evidence of cell loss (as opposed to cell arrest as p21 is later shown to be upregulated with DSS treatment) assumption, is being made, based on the data provided in this section. Also line 344/352 discussion – process is claimed to be accelerated/massive exfoliation –based on what data from this paper or other citations? Line 354-361 is supposition and not in keeping with the data presented in the paper.

To address the Reviewer’s concern, we have now analyzed the epithelial surface of H&E stainings and observed a disintegrated epithelial lining at the acute colitis stage in WT mice (see Extended Data Fig. 1g) (line 129-131). This phenotype was much less pronounced in IFN- γ R KO mice. Furthermore, we stained histological sections from different stages of experimental colitis for the apoptosis marker cleaved caspase3 (C-CASP3) (see Extended Data Fig. 1h) (line 132-134). C-CASP3 positive cells on 5-days-DSS treatment were not highly abundant. However, on early DSS (3-days-DSS) treatment we captured more C-CASP3 positive epithelial cells that were being shed into the lumen and were not part of the epithelial lining anymore. Furthermore, by live imaging of organoids (see Extended Data Video 3) (line 237-239), we are now able to show that cell shedding of apoptotic cells into the lumen is increased upon IFN- γ treatment. These results are now included as Extended Data Fig. 1g-h.

In addition, we now extended our discussion and adjusted our wording to: *“...triggers increased exfoliation of terminally differentiated cells.”*. (Line 426-427)

Extended Data Fig. 1g:

Extended Data Fig. 1h:

c. Paper title & Fig 2 title – ‘Fig. 2: IFN- γ -induced removal of the differentiated compartment and transformation of progenitors into pre-regenerative Krt20+ colonocytes.’ Title needs to reflect what is shown in paper and figure. How is removal of KRT20 compartment shown exactly & how does it fit with Krt20+ cells being increased in vivo in Fig 1?

We have now adjusted the manuscript title to *“Extrusion of BMP2+ surface colonocytes promotes stromal remodelling and tissue regeneration”* and also the chapter title to *“IFN- γ -induced KRT20+ colonocytes are different from homeostatic colonocytes”* (Line 140) and performed additional

experiments such as lineage tracing, substantial histological analysis and mouse genetic experiments (as presented in the point-by-point response to the Reviewers) to confirm our hypothesis, so that content and titles now match.

The title of Figure 2 now has been specified to: “*IFN-γ-induced colonocytes are distinct from homeostatic colonocytes.*” (Line 490)

> How are progenitors shown to be changed into Krt20 colonocytes?

We thank the reviewer for this critical question. We now performed short-term lineage tracing experiments of Axin2+ progenitors (*Axin2CreEr/Roa26TdTomato*) in assembloids upon IFN-γ treatment. The data showed that upon IFN-γ treatment there is an increased number of TdTomato positive cells that express Krt20 revealing a production of Krt20+ cells from Axin2+ progenitors upon exposure to IFN-γ (see Fig. 3h-i) (Line199-201). We also labelled progenitors with EdU and found that more EdU labelled cells were KRT20+ upon IFN-γ treatment (see Extended Data Fig. 3b) (line 204-208).

Fig. 3h-i:

Extended Data Fig. 3b:

Paper title – how is ‘programmed extrusion’ shown?

At homeostasis, crypt progenitor cells differentiate and migrate towards the intestinal lumen. When reaching the luminal surface, the differentiated cells die and are shed into the lumen. At homeostasis this process is independent of IFN-γ. We now show that IFN-γ increases cell shedding and apoptosis (see Fig. 3d, 3j and Extended Data Video 3) (line 177-178, 208-211 and 237-239), thus, cell death and extrusion appears to be a directed process induced specifically by IFN-γ. Since “programmed” may be misleading we have changed the title and removed this word.

Fig. 3d:

Fig .3j:

Other points - Fig 2j/k – why is lineage tracing with an Lgr5-cre line not included?

The Lgr5-eGFP-IRES-CreERT2 knock-in allele is known to be selectively silenced, leading to mosaic expression of the GFP and CreERT2 proteins (Doi: [10.1016/j.stemcr.2014.05.018](https://doi.org/10.1016/j.stemcr.2014.05.018)). As both Lgr5 and Axin2 are Wnt target genes expressed in intestinal stem cells (DOI: [10.1038/s41467-019-12349-5](https://doi.org/10.1038/s41467-019-12349-5)), we used Axin2-CreERT2 expression, which is more robust and thus suitable to trace stem and progenitor cells.

The population represented by Axin2-lineage is not described.

We would like to refer to our previous paper that describes Axin2 expression in the base of colonic crypts in the stem and progenitor cells. DOI: [10.1038/s41467-019-12349-5](https://doi.org/10.1038/s41467-019-12349-5)

We have now also added the following description to the manuscript:

“In these systems, the tamoxifen-inducible Cre recombinase is either expressed under control of the stem and progenitor marker gene *Axin2*, or under control of the differentiation marker gene *Krt20*. 4-OH-tamoxifen shuttles the Cre-recombinase to the nucleus to remove the transcriptional stop cassette and enable expression of the red fluorescent protein tdTomato. Consequently, cells that derive from either *Axin2* or *Krt20* expressing cells can be detected via detecting the tdTomato fluorophore with fluorescent microscopy.” (Line 184-190)

Quantification is unclear-Krt20 cells as a % of what? DAPI+ cells? Terminology of a 50% increase in *Axin2*-traced cells is confusing, given the traced cells are presented as a %-consider altering wording.

We have specified and clarified the Reviewer’s concern in the text (% of tdTomato+ cells from all DAPI+ epithelial cells per field of view) and changed the wording to avoid any confusion. (Line 193-195, see also Fig.3)

Fig 2k&l green marker is lineage cells, not KRT20/AXIN2 protein+. Label should make this distinction clear for the reader. Top of IFN-gamma treated image is without cells-how was region selected? This does not match other images of IFN-gamma treated assembloids in Fig 2(?)

We appreciate the Reviewer's comment and have modified the labels. Fig. 3f and Fig. 3g (previous Fig.2k and 2l) are images from whole-mount staining and a representative image is displayed. The focus here is on the proportion of traced cells within the remaining epithelial layer. We assume that shed cells are not visible in this particular image, as they have detached from the epithelium and were distributed into the hollow lumen of the assembloid. However, to further support our finding, we have now performed IF staining in those *Axin2* and *Krt20* lineage tracing assembloids for traced cells, KRT20, C-CASP3, and E-cadherin. (see Fig. 3h and 3j) (Line 199-211)

It appears as though the *Axin2*-lineage cells are likely the predominant epithelial population following IFN-gamma treated but this is not detailed in the summary for this section, only the KRT20 cells? How does this result (and the reduced KRT20 IF in 2h +IFN-gamma) fit with the *in vivo* increase in KRT20+ population following DSS treatment?

We would like to clarify this issue. The result of the tracing experiment does not show the actual number of *Axin2*+ cells but rather the cell population of *Axin2*+ cell-derived progeny. Thus, the data reveal that upon IFN- γ treatment the cell population derived from *Axin2*+ cells is increased. In this particular experiment, we did not analyze the cell phenotype of the *Axin2* progeny. In contrast, the homeostatic *Krt20*+ cells do not produce any progeny and their number is reduced. The data fit well with the *in vivo* data, and together with the *in vitro* organoid data allow us to conclude that already differentiated cells are lost, while new *Krt20*+ cells that derive from *Axin2*+ progenitors are produced.

What does box plot denote for this? 25th/75th percentile? Is n=5 from biological replicates or fields of view from same experiment?

We decided to use a box plot to give a quick overview on the variability of the data. n=5 refers to 5 biological replicates (each data point from an individual assembloid system). This has now been added to the legend. (Line 512)

Line 191/199 – stated that the organoid and assembloid data ‘revealed the *de novo* production of colonocytes’ How? Ki67 goes down, so lineage alteration (or simply colonocytes that don’t express *Bmp2*) rather than ‘*de novo* production’?? Need to be more careful with terminology. As written it is confusing for the reader.

We are grateful for this comment and have now modified the text to avoid any confusion: “The organoid and assembloid data revealed an IFN- γ -dependent differentiation of *Krt20*+ colonocytes that

differed in their expression of other markers such as BMP2 ...". (Line 221-222).

Our data show that upon exposure to IFN- γ the newly appearing Krt20+ cells derive from the Axin2+ compartment and are an immediate progeny that are produced in higher numbers after IFN- γ treatment compared to the untreated condition. To substantiate this finding, we have now performed lineage tracing as outlined below.

Furthermore, we show that colonocytes that differentiated in response to IFN- γ escape the IFN- γ -induced cell death, which is otherwise induced in differentiated colonocytes. Thus, after the initially accelerated cell-turnover, the IFN- γ -induced cell cycle arrest combined with the stabilized colonocytes halts the turnover and prevents the uncontrolled cell loss observed in IBD patients (<https://doi.org/10.1177/0300985814559>, <https://doi.org/10.1111/j.1749-6632.2012.06523.x>).

To explore if the KRT20 cells present after IFN- γ treatment are direct progeny of Axin2+ cells, we traced Axin2 cells using colon assembloids generated from *Axin2CreERT2/tdTomato* mice and treated them with IFN- γ (See Fig. 3h-i) (line 199-201) The data revealed an increase in KRT20+ cells that are also positive for the lineage tracing marker upon IFN- γ treatment. Thus, IFN- γ induces the generation of new Krt20+ cells.

Fig.3h-i:

d. Fig 5 does not functionally remove BMP2 signalling and yet makes claims about 'loss of BMP2' or 'BMP signalling' (also discussion line 301, 337, 341 etc). Overstatement currently, but fairly straightforward to provide more data on this point. Line 255/264 –'loss of BMP2' can be more effectively modelled using Bmp receptor knockout lines or addition of BMP antagonists to support this claim, rather than exogenous additional of Bmp2 ligand. Signaling can be examined by Smad deletion etc. Lack of addition of ligand does not always equate to situations when signaling is 'lost'.

We appreciate the Reviewer's suggestion and have now generated *Col1a2Cre/Alk3* mice and analyzed the expression of Hgf, comparing WT and *Col1a2Cre/Alk3* mice. The latter express the Cre-recombinase in the stromal compartment, which is consequently deficient the BMP receptor *Alk3* (Line 301-307, Fig. 6j). Thus, activation of BMP-signalling in stromal cells is disrupted. Indeed, intestinal stromal cells deficient for BMP signalling exhibited a strong increase in Hgf expression, which was accompanied by Ki67 and thus indicative of proliferative surface colonocytes. The data support the finding that HGF participates in activation of proliferation. We further substantiated this finding by interfering with BMP signalling in primary stromal cells, which resulted in upregulation of Hgf (Line 300-301, Fig. 6i).

We integrated this data as Fig. 6i-j

Fig. 6i:

Fig. 6j:

e. Evidence for 'extrusion'/death of colonocytes? Fig 3c – where is proof that cells are dead but Fig 3d cells are not, given caspase-3 stain in other sections of paper & claims that cells 'die' vs live? Line 299 'inducing apoptosis of BMP-2 expressing colonocytes'-what is the evidence presented for this? Softer wording or provide data to support.

Our data reveal a selective cell death induction by IFN- γ (see Fig. 4c-e) (line 232-237). To further corroborate this, we performed live cell imaging of organoids treated with IFN- γ while labelling apoptotic cells. The data show that shedding of apoptotic cells is increased upon IFN- γ . We incorporated this data to Extended Data Video 3 (line 237-239).

We have also softened the wording, as suggested.

Fig. 4c-e:

f. Line 293 – ‘regeneration that is independent of Lgr5+ stem cells’ and Line 304 in discussion. Yes, this is likely given the body of literature on this subject, however, the data presented do not present information about dependency on the Lgr5 stem cell population, only Lgr5 marker levels. To make this claim would require lineage tracing or similar technique (Lgr5 ablation) to examine contribution from Lgr5 cells. Re-word or provide new data.

We have now changed this sentence to: “Effectively, this regulatory mechanism enables epithelial reprogramming into the regenerative state from an IFN- γ -induced colonocyte state and explains how non-stem cells contribute to regeneration” (Line 354-356)

g. Discussion Line 313 ‘orchestrated endogenous process that extends to undamaged tissue regions...’ where are ‘undamaged’ tissue regions included in the data? Overstatement? Ext data 4d is not referred to in the text and contains an ‘ulcerated’ region of colitis. Please provide high resolution H&E or other image to show region of ulcer and describe results and conclusions in text or remove.

We thank the reviewer for their careful reading. To avoid any confusion, we now have removed the data from the manuscript.

h. Discussion line 322 – ‘KRT20+ cells... can be reprogrammed to proliferate and enter the regenerative state’ Over-claim, please re-word to show assumptions/suggests etc. The data do not show this – there is one lineage tracing experiment that does not show KRT20+ precursor give rise to proliferative cells.

i. Line 324 ‘...do not express BMP-2, are unresponsive to BMP signaling’ There remains some (ie not none) Bmp2 expression in Fig 2a colitis in WT and organoids +IFN-gamma 2e-temper wording.

In response to comments by all Reviewers we performed further experiments that provide evidence that IFN- γ induced colonocytes harbour regenerative capacity and are shifted into proliferation by HGF. However, we rephrased this part of the discussion part to avoid overstating the facts: “Our data reveal a KRT20+ cell population that is induced and enriched by IFN- γ upon injury. This cell population is distinct from homeostatic, terminally differentiated KRT20+ colonocytes, harbours regenerative capacity and can regain proliferative activity. These IFN- γ -induced KRT20+ cells are characterized by reduced expression of BMP-2 and reduced BMP responsiveness. Further, they are cell cycle arrested while exhibiting nuclear YAP. Nuclear YAP is a hallmark of the regenerative cell state⁴⁵ and appears inconsistent with the IFN- γ -induced cell cycle arrest. However, our finding correlates with a previous report by Nava et al. of activated nuclear β -catenin in IFN- γ -treated and cell cycle arrested intestinal cells⁴⁶. They suggested that IFN- γ induces a biphasic proliferative and apoptotic response in intestinal epithelial cells with an initial increased proliferation followed by loss of proliferative activity and apoptosis despite continued β -catenin activation⁴⁶. Together with our data, this implies that IFN- γ shifts the cells towards a colonocyte state that preserves the capacity for regeneration.” (Line 373-384)

2. At the molecular level, can additional information be provided on how HGF is upregulated in

response to IFN-gamma treatment? How does HGF then result in release of epithelial cells from arrest?

We are grateful for this critical question, which we have now addressed. The resulting data has strongly improved our manuscript. We explored whether the upregulation of Hgf in the stromal compartment is induced by the reduction in epithelial BMP production and reduced BMP-signalling activation in the stromal compartment. We now also provide new in vivo data and demonstrate that conditional knockout of the BMP-receptor in the stromal cells (now in Fig. 6j), caused a strong upregulation of Hgf. We also have performed functional experiments using primary stromal cells treated with IFN- γ and/or BMP-2 (Fig. 6f, g and i). In summary, our data now provide mechanistic explanation for the upregulation of Hgf in response to IFN- γ , which shows that loss of BMP-2 signalling is an important prerequisite for the increased HGF production (line 290-307).

The pro-proliferative effect of HGF has been described before. We now refer to the most relevant publications that demonstrate the ability of HGF to induce cell cycle progression and cell division of a variety of epithelia as well as endothelial cells (doi.org/10.1038/nrm1261, doi.org/10.1038/nrc3205). It has been shown that HGF controls both the cell cycle transition from G 0 to G 1 as well as the entry into and progression through S phase (Doi: [10.1073/pnas.0403412101](https://doi.org/10.1073/pnas.0403412101)). In essence HGF signaling controls the exit from quiescence cell state. (line 410-413)

3. Discussion should include previous literature on the role of BMP4/BMP antagonists in regeneration (eg PMID: 33819486)

We thank the reviewer for highlighting this important study, which we have now incorporated in our discussion: *“Leedham and collaborators analyzed the role of BMP signaling in intestinal regeneration and showed a reduction of epithelial BMP-4 production and an increase in stromal GREM-1 production during colitis. They demonstrated that the suppression of BMP signaling by stromal upregulation of the BMP antagonist GREM-1 protected from ulcerative lesions upon DSS-mediated colitis induction. The data highlight that suppression of BMP signaling is critical for intestinal regeneration⁴⁸.”* (Line 392-397)

Minor points

1. scRNAseq analyses. Line 143/4– Fig 2b has no statistical evaluation of differential expression of Krt20 & Bmp2 included for the reader- depict as violin plots or similar with appropriate statistical evaluation. Line 149/50 - Fig 2e – no statistically sig difference in to decrease (including Ki67 which fits with subsequent staining image). Line 195 ‘differentiation markers were increased’-needs stats. Fig 5h, line 263/4 needs correlation analysis.

We thank the reviewer for these comments and have now included a violin plot and a data table for appropriate statistical evaluation of the data (Extended Data Fig. 2e and Extended Data Table 1) (Line 153-156)

Extended Data Fig. 2e:

Extended Data Table 1:

Corresponding p-values

Celltype	GeneSymbol	p_val	avg_log2FC	pct.1	pct.2	p_val_adj
Colonocytes	Bmp2	1.063813E-08	-1.0327533	0.277	0.592	0.0002419535
SC	Bmp2	1.801412E-17	-1.4318483	0.03	0.15	4.097131E-13
Secretory	Bmp2	1.346806E-10	-1.2795962	0.038	0.256	3.063175E-06
TA	Bmp2	1.184192E-19	-0.9140337	0.112	0.34	2.693326E-15
Colonocytes	Krt20	0.3164751	0.9638114	0.375	0.374	1
SC	Krt20	0.8088588	0.6159561	0.02	0.018	1
Secretory	Krt20	0.0007944942	-2.5921778	0.023	0.098	1
TA	Krt20	0.7998547	1.1080759	0.037	0.035	1

2. Fig 2d-intensity of DAPI and E-cad staining is not consistent across all panels

We have now double checked our data and aligned the intensity of the DAPI-channel.

Fig.2d:

3. Line 252- need to explain that the 'IFN-gamma response' is based on transcriptomic measurement using a scoring metric.

We have now adjusted the text accordingly: "Those stromal cell populations (fibroblasts 1 and 2) that upregulated *Hgf* most strongly also showed the strongest IFN- γ response in the context of colitis based on GSVA analysis." (Line 287-288)

4. Fig 3a, d n=? Fig 3e – n=3-is that independent expts with same colon organoid line? Error bars denote?

Fig. 3a has been changed to Fig. 4a; n=3. Fig. 3c-e have been changed to Fig. 4c-f: we used 4 biological replicates; thus 4 colon organoids line each generated from a different mouse. We have added these informations to the figure legend. (Line 525 and 533)

5. Fig 4 legend title – Induction of [partial? ie not proliferation?] regenerative features.

We thank the Reviewer for this suggestion and have now changed the legend title accordingly using "partial". Fig. 4 has been changed to Fig. 5. (Line 537)

6. Fig 5a n=30 – organoids? Independent replicates or all in one expt? What do dashed lines denote in violin plots? Fig 5e n=3 – mice? Error bars denote?

Fig 5a has been changed to Fig. 6a, n=30 from 3 biological replicates. In each replicate, the 10 biggest organoids were used for quantification. Dashed lines at both ends indicate 25th and 75th percentiles.

A dashed line at the median indicates the 50th percentile. Fig 5e has been changed to Fig. 6e, n=3 means 3 mice. Error bars denote standard deviation. This information has now been added to the figure legend. (Line 552, 558 and 572).

7. Ext data fig 2a n=3 means 3 independent cultures of the same organoid line? 2e-legend but IF of p21 in figure is missing and not mentioned in text.

Extended Data Fig. 2a was changed to Extended Data Fig. 2c. n=3 means 3 independent colon organoid lines, each generated from a different mouse. This information has now been added to the figure legend. (Line 616)

We have removed the description of the previous Extended Data Fig.2e from the figure legend.

8. Ext data fig 4 – IL22 not expressed or below QC cutoffs?

IL22 expression was below the QC cutoffs. Extended Data Fig. 4a was changed to Extended Data Fig. 6a.

9. Line 324 typo ‘homoeostatic’

This has now been corrected. (Line 374)

Reviewer #2 (Remarks to the Author):

Heuberger et al. investigate the role of IFN- γ -dependent inflammatory response in colonic epithelial injury. The authors induced epithelial damage by exposing IFN γ R knockout and wild type mice to DSS to study the effects on the present cell types and marker expression. In an acute phase of injury, the authors find a surprising IFN γ R-dependent downregulation of proliferation and stem cell markers and an increase of KRT20+ colonocytes. Experiments in colon organoids and assembloids containing both colonic epithelial and stromal cells revealed an induction of apoptosis and reduction of proliferation upon IFN γ treatment. Using scRNA seq, IHC, IF, and qPCR the authors find a reduction in stem and secretory cell identity and an increase of a KRT20+ colonocyte type with reduced BMP-2 expression. Using a lineage tracing approach, the authors argue that IFN γ treatment promotes the loss of differentiated cells and the de novo generation of these KRT20+, BMP-2-low colonocyte that are in cell cycle arrest (“pre-regenerative colonocytes”). According to the authors’ model, the prevalence of this transient cell type causes a re-organization of the mesenchymal niche stimulating the latter to express HGF, which in turn releases pre-regenerative colonocytes from cell cycle arrest and induce proliferation to promote tissue regeneration.

The findings reveal a complex interaction between epithelial and mesenchymal cells in response to injury-caused inflammation and a surprising transient pre-regenerative state at an initial injury phase. The study is well-conducted and follows a logical experimental design supporting the proposed model. The results of this study are novel and will be of interest for a broad audience and suitable for publication In Nature Comm.

There are a few points the authors should address to further validate the proposed model:

The scRNA sequencing was done on organoids in expansion medium. The authors state that in expansion medium, differentiated cells are not present. Yet, they analyze different (mature?) cell types in this medium. Why was this analysis not performed e.g. in assembloid samples that contain the relevant cell types (stem, TA cells, colonocytes, etc.)? It is unclear how well the data characterizes the behavior of mature cell types (as suggested by the applied labels) or rather only apply to progenitor cells. Clarification from the authors would be appreciated.

We thank the Reviewer for highlighting this issue. The main proportion of cells in organoids grown in expansion medium are progenitors and only a small cell population represent differentiated cells. Although the number of cells is low, the data further substantiate our hypothesis that Krt20+ cells are different upon IFN- γ treatment. We performed various analyses to validate the findings.

Additionally, the scRNA seq seems to show that all cell types and multiple genes are affected by IFN γ treatment. It would be good if the authors could clarify their selection criteria/rational for their further focused investigation e.g. on BMP-2.

The rationale for setting the focus on BMP-2 results from the observed alterations in the Krt20+ cell compartment, which also expresses BMP-2. Furthermore, Parikh et al. observed the highest absolute values in the colonocyte lineage comparing healthy and inflamed tissue. In addition, the number of differentially expressed genes across subpopulations is high in colonocytes (data not included in manuscript).

The UMAP of both, control and IFN γ -treated organoid sample, should be shown. Does the colonocyte population shift/separate more apart or closer relative to the progenitor population?

We have now generated comparable UMAP plots that show upon IFN- γ treatment a closer shift of the colonocyte population to the stem-cell/TA2 cluster (See Extended Data Fig. 2a-b) (Line 142)

Extended Data Fig. 2a:

Extended Data Fig 2b:

In general, the manuscript would benefit from a deeper characterization of the identified transient KRT20+, BMP-2-low colonocyte. How do they compare to mature colonocyte and TA cells, respectively? I would suggest profiling these cells at the transcriptomic level using scRNAseq ideally from tissue, which should be well possible given their high abundance upon DSS treatment, or alternatively from assembloids upon IFN γ treatment. Are there any other specific markers that characterize them, and do they express other genes that would support their proposed role in reprogramming the mesenchyme?

Sc-RNAseq-analysis of colonocytes from in vivo DSS-models revealed a decreased BMP responsiveness (Extended data Fig.4a, line 228) and our lineage tracing approaches in assembloids demonstrated a clear difference in the cellular characteristics of colonocytes in the healthy state and the IFN- γ -induced colonocytes. To further characterize the IFN- γ induction we extracted the differentially expressed genes of IFN- γ and untreated colonocytes from organoids. The data reveal changes in colonocyte markers such as *Bmp2* and *Aqp8* besides inflammatory and IFN- γ related genes as such as *Cd74* and *Ly6e*. We included the the data to the manuscript (Extended data table 2, line 158-160)

he authors suggest that upon IFN γ treatment differentiated cells are lost and colonocytes form de novo from stem cells that are locked in cell-cycle arrest until upon HGF stimulation the tissue enters the regeneration phase and cells proliferate. The authors seem to suggest that it is specifically this population of colonocytes that de-differentiates and regenerates injured tissue rather than remaining stem or TA cells (that might be reduced in number but sustain upon IFN γ treatment or injury-induced inflammation). To validate this hypothesis, the authors should show more directly that it is the KRT20+ (BMP-low) colonocytes that proliferate upon HGF exposure. A co-staining of Ki67 and KRT20 would be essential.

We have now performed co-staining of Ki67 and KRT20 of control and IFN- γ treated organoids, which revealed an increase in double positive cells upon HGF treatment (Data incorporated as Fig. 7g) (Line 338-341). To explore if the KRT20 cells present after IFN- γ treatment are direct progeny of Axin2+ cells, we now have performed lineage tracing and traced Axin2 cells using colon assembloids generated from *Axin2CreERT2/tdTomato* mice and treated them with IFN- γ . These experiments revealed an increase in KRT20+ cells that are also positive for the lineage tracing marker, indicating that their origin is the Axin2+ cell population. Thus, IFN- γ fosters the generation of new KRT20+ cells (Data incorporated as Fig.3h, see Line 199-201 / see also Reviewer #1). Furthermore, we performed a KRT20+ tracing of control and IFN- γ treated assembloids, these data demonstrated that homeostatic Krt20+ cells are lost upon exposure to IFN- γ

To explore whether the IFN- γ induced KRT20+-cell-state has an increased potential to regenerate in comparison to normal homeostatic KRT20+ cells, we compared the reseeding capacity of KRT20+ cells from assembloids untreated or treated with IFN- γ . To do so, we disintegrated IFN- γ treated and control assembloids and seeded single cells into classical organoid cultures. To determine if the IFN- γ induced KRT20+ cell state has an increased potential for regeneration, we traced the KRT20+ expressing cells to monitor their direct progeny. The data revealed an increased regrowth of KRT20+

cells in the IFN- γ condition, now shown as Extended Data Fig.7 d-e. (Line 333-337)

Fig.7g:

Extended Data Fig .7d:

Extended Data Fig .7e:

The authors should design an experiment to test the hypothesis at the functional level. They could make use of their lineage tracing models to test if de novo formed KRT20+ colonocytes replicate to form new tissue after HGF treatment or rather the remaining SCs. Can these 'pre-regenerative colonocytes' grow entire organoids from single cells upon HGF treatment?

We thank the reviewer for this great suggestion. Following their advice, we have now treated organoids with IFN- γ and traced then KRT20 expressing cells by reseeding them as a single cell suspension into full medium without IFN- γ . The subsequent seeding experiment revealed that indeed more KRT20-traced organoids regrew when co-treated with HGF, and Ki67 staining confirmed that the KRT20-traced cells gained proliferative activity.

We show these data in (Extended Data Fig.7d-f), see also line 333-338: "To further explore if HGF promotes regrowth of IFN- γ -induced colonocytes, we treated *Krt20CreEr-tdTomato* organoids with IFN- γ and explored the colony-forming capacity of the KRT20 lineage. First, we confirmed a higher abundance of KRT20+ cells upon IFN- γ treatment compared to untreated controls (Extended Data Fig.

7d). Single-cell seeding into expansion medium showed a higher regrowth rate of tdTomato+ cells in the HGF-treated condition (Extended Data Fig. 7e), and those KRT20 traced cells expressed the proliferative marker Ki67 (Extended Data Fig.7f).”

Extended Data Fig.7d:

Extended Data Fig.7e:

Extended Data Fig.7f:

Do colonocytes dedifferentiate into S/TA cells and lose KRT20 upon HGF treatment? A KRT20 staining upon HGF treatment would be great.

Interferon-induced colonocytes can be dedifferentiated into S/TA cells according to our KRT20 tracing data in Extended Data Fig. 7f (line338). We have now reseeded those cells with KRT20 tracing into

FM after IFN- γ or IFN- γ + HGF treatment. We found that those KRT20 traced cells became Ki67+ cells. We did not observe Krt20 decreasing upon HGF treatment when co-treated with IFN- γ according to our qPCR data in Extended Data Fig. 7b (line 322-324). However, we observed increased numbers of KRT20+ and Ki67+ cells upon HGF + IFN- γ treatment (see Fig. 7g, line 329-330). Taken together, those data demonstrate that HGF does not interfere with Krt20 expression but increases their proliferative activities.

Extended Data Fig. 7f:

Extended Data Fig. 7b

Fig. 7g:

Reviewer #3 (Remarks to the Author):

Dextran sodium sulphate (DSS) treatment is a popular model for colonic epithelial injury. The authors of this manuscript previously reported a role for IFN γ R in a gastric infection model (Kapalczyńska et al, 2022) and here they extend the theme to DSS-injured colon, though much of the study hinges on IFN γ -treated colonic organoids. They propose a model where IFN γ simultaneously promotes loss of BMP2-expressing colonocytes, cell cycle arrest, reduced stem cells, and increased differentiation marker Krt20. These cells are proposed to be “pre-regenerative” (neither stem-like nor differentiated). Upon loss of BMP2-expressing colonocytes, the underlying mesenchyme also decreases BMP and increases HGF signaling. HGF overrides IFN γ -induced effects on organoids, promoting return to epithelial homeostasis following injury.

Major comments:

1. The authors go back and forth between DSS colitis in vivo and IFN γ -treated organoids. Even if there are some parallels, DSS colitis is clearly different from an IFN γ response, and many in vitro and in vivo findings don't align, e.g., secretory (goblet) cells are well preserved in colitis (Ext. Fig. 1a-b) but

significantly compromised in IFN γ -treated organoids (Fig. 2b). A conflation of experimental models leads to a convoluted model of unusual phenomena (death and production of selected cell types) that require more rigorous evaluation before being ready for publication.

We appreciated the Reviewer's point. However, we did not aim to explore the mechanism underlying DSS-colitis with the effect of IFN- γ , but rather used it as a model to dissect the role of IFN- γ during the processes of regeneration. To this end, we used in vivo genetic models to explore the role of IFN- γ signaling by comparing the epithelial responses to injury between WT and IFN- γ KO mice as well as organoids and assembloids to directly explore cellular responses to this specific cytokine.

We re-evaluated our data shown in Extended Data Fig. 1a-b, Fig.2b and Extended Data Fig. 2c. (Line 98-101, 148-149 and 155) The reduction in secretory goblet cells as demonstrated by marker gene reduction (Fig. 2b and Extended Data Fig. 2c) is also observed in in vivo. Extended Data Fig. 1a on the left shows the untreated controls of WT and IFN γ R KO mice. Extended Data Fig. 1a on the right side shows the DSS-treated condition of WT (upper) and IFN γ R KO (lower) mice. The WT mice treated with DSS clearly show a reduction of goblet cells. We have now added a quantification of goblet cells to the figure, which confirms the reduction in goblet cells (Extended Data Fig. 1a, on the right). The quantification of the number of enteroendocrine cells also revealed a trend towards reduced cell numbers upon DSS treatment (Extended Data Fig. 1b, on the right).

Extended Data Fig. 1a:

Extended Data Fig. 1b:

Fig. 2b:

Extended Data Fig. 2c:

2. The authors suggest that IFN γ signaling causes rapid colonocyte loss after DSS treatment. The claim relies on a finding that *Bmp2* expression is reduced in acute DSS colitis in WT but not in IFN γ R KO mice (Fig. 3a-b, not quantified and not entirely convincing). The authors do not distinguish between expulsion of *Bmp2*-expressing cells (their interpretation) and the simpler explanation of reduced *Bmp2* expression. Experiments in Fig. 3c-d do not distinguish between these possibilities. There are few apoptotic cells and the manuscript presents no evidence for “programmed extrusion of BMP2+ surface colonocytes” (title, etc.). In fact, *Krt20*+ cells are increased, not decreased, in acute colitis (Fig. 1e). Additional and better studies are needed to support the conclusion that “BMP-2 expressing homeostatic colonocytes die in response to IFN- γ , while IFN- γ induced de-novo colonocytes do not”.

We thank the Reviewer for these critical comments as they helped us to make improvements to the manuscript, particularly to the description and explanations of our findings.

We agree that there are two potential explanations of the phenotype: expulsion of *Bmp2* expressing cells or reduction of *Bmp2* expression.

We started from observations: increase of *Krt20*+ expression (Fig.1e-f) and reduction of *Bmp2* expression (Fig.4a-b). Both effects are not present in the IFN- γ R KO mice, which reinforced the notion that this effect is mediated by IFN- γ . To decipher this in more detail, we have now performed organoid experiments (Fig. 1j, Extended Data Video 1, Extended Data Fig. 1d-e) revealing that IFN- γ induces cell death of differentiated cells but not of progenitor cells. Lineage tracing in assembloids (Fig. 3f and 3j) revealed that primarily differentiated *Krt20*+ cells die in response to IFN- γ . We were thus surprised that *Krt20* expression increased in response to IFN- γ (Fig. 2b and 2e, Extended Data Fig. 2e). One explanation could be that *Krt20*+ cells that differentiated during healthy conditions (“old *Krt20*+”) express *Bmp2* and are highly sensible to IFN- γ , while “de novo *Krt20*+” cells formed under the influence of IFN- γ , have reduced BMP signal expression and are less sensible towards IFN- γ .

Therefore, we performed lineage tracing of Krt20+ and Axin2 (progenitor) cells (Fig. 3f-j) (see line 194-216) which demonstrated that in response to IFN- γ the number of “old Krt20+” cells was reduced, while the number of “de novo Krt20+” cells (direct Axin2 progeny) was increased. These “de novo Krt20+” cells do not exhibit all features of mature colonocytes, e.g. show a reduced expression of Bmp2.

BMP2 is a known driver of colonocyte differentiation and is expressed by differentiated colonocytes itself. Thus, terminal differentiation is fostered by an BMP feed-forward loop (doi: 10.1038/s41467-022-29176-w.) and BMP can induce colonocytes differentiation of neighbouring cells. To delineate whether the reduction of the BMP-2 signalling circuitry is causative for the IFN- γ response of colonocytes we performed now experiments with sequential treatments of BMP-2 and IFN- γ . We enforced differentiation by treating progenitor organoids first with BMP-2 followed by IFN- γ treatment (Fig. 4c-d) (line 232-236). BMP-2-treated and BMP-2-dependent differentiated organoids die in response to IFN- γ . Vice versa, progenitor organoids treated first with IFN- γ and subsequently with BMP-2 showed a different response (Fig. 4 c-d). IFN- γ treated and IFN- γ -dependent differentiated organoids remain stable in response to BMP-2. Marker genes expression analysis (Fig. 4f and Extended Data Fig. 4b) revealed that IFN- γ reduces Bmp2 expression as well as BMP-2 responsiveness (Fig. 4f) (see line 239-240) which we further confirmed by sc-RNAseq analysis (Extended data Fig. 4a) (line 227-228).

3. A key element of the model is that IFN γ -treated colonocytes represent a “new” state, with simultaneous features of proliferation and cell cycle arrest (e.g., cell cycle marker Ccnd1 is increased but Mki 67 is decreased). These and other interpretations are needlessly far-fetched. A simpler explanation is that scRNA-seq data are technically limited (Fig. 2b – e.g., cell proportions from untreated and IFN γ conditions are not indicated) and interpreted too liberally.

As described above we have now performed lineage tracing analyses in assembloids and demonstrate that indeed IFN- γ induces progenitor cell differentiation into Krt20+ cells, while homeostatic “old” Krt20+ cells die in response to IFN- γ .

Furthermore, to address the appearance of a “new” KRT20 cell state, we performed co-staining of Ki67 and KRT20 of control and IFN- γ treated organoids, which revealed an increase in double positive cells upon HGF treatment (shown as Fig. 7g). (Line 328-330) To explore if the KRT20 cells present are direct progeny of Axin2+ cells after IFN- γ treatment, we now traced Axin2 cells using colon assembloids generated from *Axin2CreERT2/tdTomato* mice and treated them with IFN- γ . The data revealed an increase in KRT20+ cells that are also positive for the lineage tracing marker upon IFN- γ treatment. Thus, IFN- γ fosters the generation of new KRT20+ cells (shown as Fig. 3h-i, line 199-201)/ see also answers to Reviewer #1 and #2). Furthermore, we performed KRT20+ lineage tracing of control and IFN- γ treated assembloids. To explore whether the IFN- γ induced Krt20+-cell-state has an increased potential to regenerate compared to normal Krt20+ cells, we compared the reseeding capacity of KRT20+ cells from assembloids untreated or treated with IFN- γ . To do so, we seeded dissociated cells from IFN- γ -treated and control assembloids into classical organoid cultures. To test if the IFN- γ induced KRT20+ cell state has an increased potential for regeneration, we traced the KRT20+ expressing cells that allowed us to monitor their direct progeny. The data revealed an increased regrowth of KRT20+ cells from the IFN- γ condition and thus substantiate the idea of a new KRT20+ cell state (shown as Extended Data Fig. 5f; line 268-272).

We have also added immunofluorescence co-staining of histological sections, which further substantiates the existence of a transient new colonocyte cell state that is characterized by Krt20 expression and nuclear YAP, while the proliferative activity is strongly reduced (Ki67 levels are down, p21 levels are up) in assembloids and in vivo. (shown as Fig. 5g and Extended Data Fig. 5e; line 263-268).

Fig. 7g:

Extended Data Fig. 5f:

Fig. 5g:

4. The conclusion that IFN γ induces a “pre-regenerative colonocyte state” is especially convoluted and superficial. It rests on a series of findings (Anxa1 expression, phosphorylated YAP, p21 expression) that are likely unrelated but which the authors assume to be causally related to one another. No alternative interpretation is considered and no experiments are performed to validate the “pre-regenerative state”. No evidence is presented for a presumed transition from “IFN- γ -induced pre-regenerative cells into the proliferative regenerative state”.

We thank the reviewer for these critical comments. To avoid misinterpretations, we now refer to the cells as IFN- γ induced colonocytes and describe differences between homeostatic and IFN- γ induced cells.

Furthermore, we have performed functional experiments to test the regenerative capacity of these cells. To address whether the IFN- γ induced Krt20+ cell-state has an increased potential to regenerate in comparison to normal Krt20+ cells, we now compared the reseeding capacity of KRT20+ cells from assembloids untreated or treated with IFN- γ . To do so, we seeded single cells from dissociated IFN- γ -treated and control assembloids into classical organoid cultures. To test if the IFN- γ induced KRT20+ cell state has an increased potential for regeneration, we traced the KRT20+ expressing cells that allowed us to monitor their direct progeny. The data revealed an increased regrowth of KRT20+ cells from the IFN- γ condition and thus substantiate the idea of a new KRT20+ cell state (shown as Extended Data Fig. 5f; line 268-272).

Extended Data Fig.5f:

5. The connection with HGF is plausible and is supported by published data, but alternative possibilities are not considered and the model is assessed superficially. Much is made of the observation that HGF rescues the IFN- γ induced reduction of reseeding capacity when colonoids are grown in full medium but not when they are grown in differentiation medium. There are many possible reasons for that phenomenological difference but the authors consider only one. Their model is not necessarily wrong but easily could be. The necessary level of rigour (entertaining and excluding various possible explanations) is missing from the study.

We agree that indeed, HGF is likely not the only one factor that contributes to regeneration; we do not exclude the involvement of other factors in this process

We have tested other potential factors (IL-22, IL-33, NRG1) before known to participate in regeneration for their interplay with IFN- γ (see Fig. 6a, and Extended Data Fig. 6a-c) and HGF induced the strongest effect. We do not want to claim that HGF is exclusive but our data reveal that it is sufficient to cause strong proliferative response.

The reviewer raised the issue of the regrowth-inductive effect of HGF on IFN- γ treated organoids grown in differentiation medium. To address this issue, we have now differentiated organoids, treated them with IFN- γ and reseeded them. The data reveals that HGF cannot promote regrowth of homeostatic differentiated colonocytes treated with IFN- γ (shown as Fig. 7h-i, Extended data Fig. 7h-i and Extended Data Fig. 6a-c, see line 341-342 and 278-284). Furthermore, we also adjusted the discussion, naming other signalling system that also participate in regeneration (line 405-407).

Fig.7h-i:

Extended Data Fig.7h-i:

6. Do the authors propose that IFN γ decreases mesenchymal BMP2 expression? It is difficult from the data to connect epithelial BMP2 loss to the loss of BMP response/increased HGF in the mesenchyme. Could IFN γ itself not be doing both? What is the timeline of HGF expression coinciding with BMP2/Id loss in the epithelium and mesenchyme during acute DSS colitis?

We appreciate the Reviewer's comment and have now performed a set of new in vivo experiments and compared the expression of Hgf in WT and *Col1a2Cre/Alk3^{fl/fl}* mice. The latter expresses the Cre-recombinase in the stromal compartment, which is consequently deficient of ALK3, a BMP receptor. Thus, activation of BMP-signalling in stromal cells is disrupted. Indeed, intestinal stromal cells deficient for BMP signalling exhibited a strong increase in Hgf expression. These data are shown as Fig. 6j, line 301-305.

Fig. 6j:

7. Do the authors propose that HGF decreases epithelial IFN γ receptor expression (Figure 6) or a separate pathway that bypasses IFN γ -induced cell cycle arrest?

We now also analyzed the expression of IFN- γ receptors in response to HGF and observed only a mild reduction (Extended Data Fig. 7g, line 338-341). We propose that the main effect may result from a separate pathway bypassing IFN- γ signalling referring to our qPCR data of IFN- γ targets (Extended Data Fig. 7c, line 331-333).

Extended Data Fig. 7g:

Extended Data Fig. 7c:

Technical concerns:

1. Dot counts for *Lgr5* and *Axin2* (Fig. 1) do not match the images shown. Hardly any *Lgr5* dots are visible in most crypts compared to *Axin2*, yet the graphs show almost similar y-axis values.

We thank the reviewer for the critical evaluation of the data. We have re-evaluated the images and now present the most representative ones. However, *Lgr5* is only weakly expressed and thus appears only as small dots by in situ hybridization.

Fig.1a-d:

2. Colon tissue sections from experimental and control mice are not comparable in many cases (Figs. 1a, 1e, 3a-b, 4a-b). Crypt orientation is vertical in some images, tangential in others. Differences are particularly stark in Fig. 3.

We have re-evaluated our data and now present the most representative images with a more consistent crypt orientation.

Extended Data Fig. 1a:

Fig. 1e:

Fig. 4a-b (Fig. 3a-b were changed to Fig. 4a-b):

Fig. 5a-b (Fig. 4 a-b were changed to Fig. 5a-b):

3. Rudimentary phalloidin staining is not adequate to conclude that organoids treated with IFN γ have “more pronounced cell polarity.” The finding is soft and may not be relevant.

We agree with the reviewer that this finding is minor compared to our other data. However, the data further substantiate the IFN- γ -induced cell differentiation. We would therefore like to keep the data in the manuscript but have adjusted the description of the data by stating that “Actin/phalloidin staining also revealed that HGF counteracts the IFN- γ -induced cell-differentiation (Fig. 7c)” (line 325)

4. scRNA-seq. Are the authors comparing roughly similar numbers and similar quality from each condition (untreated and IFN γ treatment)? Effects of IFN γ treatment could reflect poor organoid health (e.g., Fig. 2f-g, Ext. Fig. 1d – are apoptotic markers expressed?) rather than specific Bmp2 reduction and other purported changes. The authors use >15% mitochondrial reads as a cutoff (the norm is >5%). Few markers are assessed. One could take any few scRNA markers and interpret the subsequent histological data (Fig. 2d, f, g) with a different lens. The authors should approach these data with stringent QC measures and interrogate meaningful panels of genes rather than single genes (Krt20, Bmp2) that could be misleading in sparse scRNA-seq data.

Cell ranger results of our data misclassified many droplets as cells, which could be clearly seen for cells that were located in the shoulder of background-only containing droplets in the barcode rank plot of Cell ranger. Therefore, we used a custom filter of a minimum of 5,000 unique UMIs per cell to exclude misclassified droplets. Mitochondrial reads represented higher proportions (30-50%) of cells in all conditions after this filtering step. Since those cells might represent cells affected by IFN- γ we decided to keep most of them, using a higher-than-normal threshold. Final filtering resulted in 1,058 and 917 cells for untreated and IFN- γ -treated cells, respectively. The majority of MT > 5% cells mapped to the SC subpopulation.

In a sensitivity analysis, we analysed our data with a stricter threshold of <5% mitochondrial reads, resulting in 541 and 699 cells for untreated and IFN- γ -treated cells, respectively.

The analysis according to Figure 2b after stricter filtering showed similar marker expression profiles as the more liberal analysis (see below).

5. Judging by the dots, Pbx1 seems mistakenly annotated as a progenitor marker (Fig. 2b)

We thank the reviewer for their conscientious assessment. We have now adjusted the figure to have Pbx1 annotated as a colonocyte marker.

Fig.2b:

6. Ki67 immunostaining does not look reduced in IFN-gamma-treated assembloids (Fig. 2g). Krt20 expression looks the same (possibly higher) in untreated assembloids (Fig. 2h). Assembloid data (Fig. 2g-i) are not interpreted objectively. Select and tiny quadrants are shown (other areas show less difference) and findings are not quantified. All immunostains should be quantified.

We have now quantified the respective immunostains, for which we used a least 3 independent biological replicates. Indeed, Ki67 expression was a little bit higher in IFN-gamma treated assembloids when compared to untreated assembloids. We assumed two explanations. One is that IFN-gamma kills differentiated cells so that the percentage of Ki67+ cells were increased due to the decrease of total cell numbers. The other is that IFN-gamma can also increase HGF expression in stromal cells, which might rescue the IFN-gamma effects on epithelial cells.

Fig.3b:

Fig.3c:

Fig.3d:

7. Assembloid data on cell death (Fig. 2i) are not informative (alone) because apoptotic cell burden varies widely across untreated and treated assembloids for a number of reasons. Can the authors clarify and quantify when DSS-exposure causes apoptosis in vivo? Is the effect abrogated in IFN γ KO mice? Is IFN γ expressed in colonic epithelium, mesenchyme, or both?

We have now performed live imaging of organoids (see Extended Data Video 3), which revealed that shedding of apoptotic cells into the lumen is increased upon IFN- γ treatment. Besides having another model system showing the increased levels of apoptosis induced by IFN- γ , it becomes obvious that cell shedding and apoptosis are not completely independent events and occur rather simultaneously (shown as Extended Data Fig. 1h (line132-134)). As consequence, apoptosis and cell shedding cannot be easily monitored in vivo. Nevertheless, we have now stained histological sections from different

stages of colitis for the apoptosis marker Cleaved Caspase3 (C-CASP3). Only rarely did we observe C-CASP3 positive cells during DSS treatment. At day 3 of DSS treatment, we observed C-CASP3 positive epithelial cells that were shed into the lumen and were not part of the epithelial lining anymore. Furthermore, we analyzed the epithelial surface of H&E staining and observed a disintegrated epithelial lining at the acute colitis stage in WT mice (see Extended Data Fig. 1g) (line 129-131). This phenotype was much less pronounced in IFN- γ R KO mice. In conclusion, the data from these three model systems coherently show that apoptosis and cell shedding is increased upon IFN- γ treatment.

Extended Data Fig. 1g:

Extended Data Fig. 1h:

Point-by-point response

Reviewer #1 (Remarks to the Author):

1. Overall the authors have worked to include new data to address concerns and modified the text, however concerns remain about **clarity and interpretation of findings**. The combination of the different model systems used is still not as clear as the discussion portrays, especially as discrepancies between the models are not addressed in the results or discussion section to guide the reader.

We have used the two different model systems to exploit the advantages and disadvantages of each: The classical epithelial organoids allow the analysis of direct epithelial responses, focusing on either stem cells or differentiated cells, which is useful for dissecting the responses of each cell type compartment individually. However, as organoids need to be maintained by external growth factors, it is impossible to generate conditions in which both stem and fully differentiated cells can be maintained simultaneously. By contrast, once established, assembloids do not require exogenous growth factors, as they contain stromal cells, which secrete growth factors that act directly on neighbouring epithelial cells, allowing self-organization into crypt-like structures with stem cells at the base and a full complement of differentiated cells at the top. Assembloids thus allow investigations into the response of the in vivo-like crypt as a whole, including stromal cells. We now provide more explanatory information on why we used a specific model system. See line 109 (results), and line 170-173 (results), and line 369-373 (Discussion)

At the most basic level, IFN-signalling in colitis in vivo and in vitro IFN treatment of organoids results in increased Krt20 labelled epithelia. This contrasts to IFN treatment of assembloids reported to decrease/not change percentage of Krt20+ cells.

No explanation of why results reported in the different models varies – this is glossed over which doesn't assist reader comprehension.

We thank the reviewer for highlighting this issue. Although overall, our results are highly congruent across models, the increase in the total number of Krt20+ cells seen in FM organoids was indeed not observed in assembloids. It is evident that both organoids and assembloids exhibit a rapid reduction of homeostatic Krt20+ cells upon IFN- γ treatment. Krt20+ cells that are present after treatment represent de-novo Krt20+ cells, which is also consistent across models. The absolute number of cells in assembloids is not changed significantly because there is already a high number of homeostatic Krt20+ cells at baseline that are replaced by de-novo Krt20+ cells. We now discuss this and provide an explanation of the differential abundance of Krt20+ cells in the different model systems (see line 369-372 and 390-393 (Discussion)). Furthermore, the discussion identifies the key roles of IFN- γ . However, it is highly likely that in vivo IFN- γ is not the only inflammatory mediator affecting cellular extrusion and differentiation. Thus, our data offer explanations for a single key process, which is likely to act in concert with others. We hope that this puts the data into the right context and helps to avoid overstatements.

Extrusion of BMP2+ surface colonocytes is still a main focus of the paper but some of the data is unconvincing/conclusions over-stated.

We have checked the entire text and removed statements that may represent overstatement. For instance, we altered our conclusion of the data presented in Fig 5 (former Fig.6) to:

“This suggests that the loss of BMP signaling may allow for the remodeling of the stroma and increased production of HGF after injury.” (line 320-322).

Or: “IFN- γ signaling participates in regenerative state induction in vivo” (line 256, heading)

Or : “These data suggest that IFN- γ is involved in the reduction of Bmp2-expressing colonocytes and in the enrichment of de-novo differentiated colonocytes that neither express Bmp2 nor have active BMP signaling.” (Line 242-244)

2. Wording over-stated, results images not consistent across manuscript.

a. Line 131 'indicating an increased IFN-g dependent loss of mature colonocytes in acute colitis'

The WT colitis H&E representative image included in Ext data Fig 1g doesn't match the other images included with different stains from this same group in the paper (eg luminal surface Fig 1a/1e-not such obvious 'disintegration'). How then is this image representative?

This and apoptotic stains are key in vivo data in the paper showing potential INF dependency of the apoptosis/disintegration phenotype. While position of cells is consistent with KRT20+/Bmp2+ colonocytes, suggest remove this claim/re-word unless co-stain of TUNEL or CASP3 with KRT20+/Bmp2 can be provided – presumably 3d DSS timepoint captures this best, rather than d5 (no quantification provided for Ext data fig 1h).

We aimed to guide the reader by the images and not only by the quantification. For that reason, we had chosen an image that clearly shows the disintegration. However, we agree with the reviewer and exchanged the image to a more representative one.

We also tempered our wording (line 133-135) and re-worded statement (exchanged “loss” with “reduction”, line 140).

As Casp3 was very hard to detect we were not able to quantify the number of positive cells. For this reason, we also used TUNEL staining to confirm the alteration in the number of apoptotic cells.

b. Line 176-177 – quantification is now included but is not consistent with the statements in the text – '...IFN-g...reduced numbers of Ki67+...decrease in KRT20+' when quantification now shows Ki67 is increased with IFN-g and KRT20+ not sig different? Is text referring to a specific region not quantified? Unclear how assembloid regions of interest were selected, prone to bias in selection of small region from the large assembloid. Zoomed out versions of the assembloid would be informative for the reader.

We have now clarified our findings in the manuscript, and provide overview images in extended Fig 3a. In this experiment we observe the appearance of cells that are double positive for Ki67+ and KRT20+ upon IFN- γ treatment and now also provide quantification. By contrast, in controls those markers distinguish proliferative (Ki67+) and differentiated (KRT20+) cells. We do not observe changes in net KRT20+ cell number upon quantification. However, the observation of a Ki67+/KRT20+ double positive cell population, in light of the lineage tracing data, reinforces the notion of the de-novo appearance of these cells.

c. Fig 3f IFN treated assembloid image does not match any of the other similarly treated assembloid images -authors previously responded that they presumed the shed cells had detached in this image. But now with new images, especially new fig 3h with KRT20 IF, 3j KRT20 tracing in apoptotic cells - why is a region with cells in only half the image chosen for 3f, especially when major conclusion is about 'extrusion' of KRT20+ cells? Are additional timepoints required to capture? No mention of why ACTIN is used for costain in images for the reader (not mentioned in text or figure legend)

The difference between the images in Fig. 3f (now Fig. 2g) and the other images is explained by the fact that the images of Fig. 3f (now Fig. 2g) were generated by whole-mount imaging and not from sections. For this, we positioned the assembloid for imaging and acquired images by randomly picking areas of the assembloid. We selected this particular image of the IFN- γ treatment condition as it shows the flattening of the crypt structure in a representative manner. Using wholemounts for this experiment is also the reason why actin/phalloidin served as a co-stain to visualise the cell boundaries.

We now provide this information in the text (line 198) and in the figure legend.

d. Ext data video 3 is unconvincing – no quantification or indication of replicate numbers in figure legend.

We have now added quantifications of the shed cells to the Extended Data Fig.4 b.

Line 137 'a loss of mature colonocytes' but apoptotic cell death was by the authors admission 'rather low' and KRT20 cells are increased with colitis, so this statement is misleading/one-sided as written.

We believe that the selective increase of de novo KRT20+ cells while mature KRT20+ are decreased was adequately addressed experimentally with the organoid and assembloid models. However, we have now modified the conclusion sentence (line 140) to avoid one-sided conclusions.

e. 7 figures not warranted

- Fig 2 could be moved to extended data to reduce figure number in manuscript proper

To follow the reviewer's suggestion we now moved Fig. 2 to the extended data Fig. 2.

f. Fig 3 - The conclusion from this section does not accurately represent the data, it is very one-sided and overplays the lineage tracing – these are ~2d tracing studies only, so many of the positive cells will be Krt20 or Axin2 expressing rather than progeny. Eg text states 'promoting differentiation of Axin2 progenitors into Krt20+ colonocytes' & 'promotes the generation of new KRT20+ cells' when overall the number of KRT20+ cells is not different with IFN treatment (Fig 3c quantification n.s.), the number of KRT20 lineage traced cells decreases with IFN and doesn't acknowledge the key increase in Axin2 'progenitor' type cells in the culture with IFN treatment. This flows through to the summary statement at the beginning of Fig 4 text 'The organoid and assembloid data revealed an IFN-dependent differentiation of KRT20+ colonocytes'. The organoid data shows a partial differentiation phenotype. The assembloid data does not support a differentiation increase – apoptosis, crypt reduction, possible increase in Wnt+/Ki67+ population yes but not increased differentiation from data presented.

The statements have now been revised to include additional explanatory information about the hypothesis, the tracing experiment and the results. In addition, the wording has been adapted to avoid any overstatements. Furthermore, we adopted the reviewer's suggestion on "partial differentiation". Thus, we changed the paragraph referring to Fig. 3 (line 178-189, 198-201 and 209-209) and the summary statement at the beginning of Fig.4 (line 233-234) (now Fig. 2 and 3 respectively).

3. Minor points

a. Line 190 – 'At 24h after induction of KRT20+ colonocyte lineage tracing' – but 3e shows that harvest point was at 54h post-OHT lineage trace induction? Alter wording.

We thank the reviewer for pointing this out. We adjusted the text to clarify the experimental time schedule (now Fig. 2f) and the correlative images (Fig. 2g-i).

b. **Fig 3j** – new data – why is this presented split into channels? The rest of the figures are not. Negative control IF for the TdTom lineage stain needs to be shown given false positive problems with dead/dying cells staining as found in centre of assembloid. Good to show no primary or similar with same secondary to confirm validity of stain.

We decided to split the channels of Fig. 2k, as it appears the most appropriate way to present the data to allow a better distinction of the signals and better distinction of the signals.

Controls are provided below for the reviewer. We selected an area in which the overlap is not as complete, which allows the signals to be distinguished more easily. We stained serial sections with a combination of primary antibodies and the entire panel of secondary antibodies. Staining in a) anti-tdTomato (traced Krt20); anti-CleavedCasp3; anti-E-cadherin and b) anti-CleavedCasp3; anti-E-cadherin.

c. **Ext data Fig 3a** – now included new analysis of a public scRNAseq dataset. Difficult to evaluate. ‘Mature colonocyte’ and ‘immature colonocyte’ labels are used, presumably based on original publication, but how were these terms assigned? On what basis? What does that mean for the markers that have been used in this paper (Krt20 in particular) & comparison to ‘progenitor/TA’/‘colonocyte’ clusters from Fig 2 scRNAseq?

We now provide a dot blot to illustrate how we defined the cell populations (including Krt20, Aqp8 etc.). See also Extended Data Fig3b. Aqp8 is normally used as a marker to cluster immature colonocytes and mature colonocytes. However, as Aqp8 is completely missing from the organoids, we needed to use Krt20 expression to visualize colonocytes. As KRT20 was used throughout our experiments as a colonocyte marker, we are indeed unable to distinguish mature and immature colonocytes from each other only based on the detected IF signal. However, using the sequencing data we were able to classify these two cell clusters.

Figure lacks standard information such as p values or other statistic to indicate significance - error bars +/-2 SEM appear overlapping? How many biological replicates, was total cell number analysed equal, noting these are absolute cell numbers rather than proportion of the total number examined?

We now provide the information in figure legend.

d. Line 308- without including analysis of DSS treated Bmpr1aKO mice, this statement has not been proven, yet wording suggests this. Modify wording to 'This suggests that loss of BMP signalling may...'

We thank the reviewer for this critical point and have altered our conclusion accordingly (line 320).

Reviewer #2 (Remarks to the Author):

I appreciate the experiments the authors conducted to validate their key results, as well as the textual revisions made to temper the boldness of their interpretations. The lineage tracing experiments help supporting the presented model.

I do find the repetitive posting of the very same results in response to the reviewers' comments irritating.

Minor problems with the new experimental data:

Figure 3h: The images and especially the color choice is not ideal. Difficult to appreciate colocalization.

We have now altered the colors (now Fig.2k)

Figure 7g: The quantification shows 40% of Ki67+ cells are Krt20+. In the zoomed image there is not a single Ki67+ cell visible but in the organoid next to the selected one, there are multiple Ki67+ cell visible, and Krt20+ is not visible at all. Did the authors choose a representative image? How heterogenous is the response of individual organoids in the IFN- γ condition, i.e. what percentage of organoids upregulate Krt20?

We thank the reviewer for pointing this out. We have now revised the data presentation and show a more representative image (NB: Fig.7 is now Fig.6). The organoids are a mixture of different cell types, so it is difficult to define the KRT20+ organoids. Therefore, we counted the percentage of KRT20+ cells.

Reviewer #2 (Additional Remarks to the Author):

The authors have sufficiently addressed Reviewer 3's concern about the use of DSS-induced colitis in WT and IFN γ R KO mice alongside IFN γ treatment in organoid models. They clarify that DSS colitis was used to induce an IFN γ -mediated injury response in vivo, while the effect of IFN γ was directly investigated by treating organoids in vitro, which I find to be a reasonable and appropriate experimental approach. The supplementary data provided on the abundance of goblet cells and enteroendocrine cells across different conditions are consistent between the experimental models, supporting the authors' interpretation of the data.

Reviewer 3 questioned the authors' interpretation regarding the loss of mature colonocytes following DSS treatment and emergence of a newly formed Bmp2^{low} expressing population, proposing instead that colonocytes may downregulate Bmp2 expression. Reviewer 3 argued that there is insufficient evidence for apoptotic extrusion of colonocytes. In response, the authors pointed out that neither an increase in Krt20+ cells nor a loss of Bmp2 was observed in IFN γ R KO mice after DSS treatment, suggesting that these effects are dependent on IFN γ signaling. To investigate further, the authors employed organoid models to examine whether Bmp2 levels decreased or if Bmp2-positive cells were lost upon IFN γ treatment. The authors introduced new lineage tracing experiments (using Axin2 and Krt20+) in organoids/assembloids and found that mature Bmp2+, Krt20+ colonocytes undergo apoptosis in immediate response to IFN γ , while a new population of immature Krt20+ cells (with low Bmp2 expression and thus reduced sensitivity to IFN γ) expands from progenitor cells. Additionally, the authors supported their findings through sequential treatments with BMP-2 and IFN- γ , in combination with qPCR, sc-RNAseq analysis, and outgrowth assays, as suggested by other reviewers. Overall, these experiments provide sufficient evidence for the

authors' hypothesis and adequately address Reviewer 3's alternative interpretation, while making its consideration clear to the reader.

The authors appropriately revised their terminology to 'IFN- γ -exposed colonocytes,' which offers a more neutral and precise description (despite 'cell state' being a less clearly defined term that could have been justified as well). They also demonstrate the proliferative capacity of these cells using an outgrowth assay. The authors should ensure they clearly indicate to the reader that their model, which proposes a causative transition from IFN- γ activation to a pre-proliferative cell state and subsequently to a proliferative cell state, is supported but not the only possible hypothesis.

Unlike Reviewer 3, I do not interpret the authors as claiming that HGF has an exclusive role in triggering proliferation. The reasoning for a more detailed investigation of HGF was well-documented, and a few other potential factors were tested beforehand. Instead, the authors use HGF to illustrate the requirement for additional triggers to initiate proliferation as part of the regenerative response. In my view, this point was clearly conveyed.

I agree with Reviewer 3 that the effects of IFN γ on the stromal compartment were not rigorously shown. Specifically, the authors do not provide evidence that IFN γ decreases mesenchymal BMP2 expression. While they did add new in vivo experiments comparing HGF expression in WT and Col1a2Cre/ALK3fl/fl mice, showing increased HGF expression upon BMP signaling inhibition, they did not perform epistasis analysis to clarify the interactions between IFN γ and BMP2, nor did they present any temporal data. More comprehensive experiments would be required to draw stronger conclusions about stromal cell regulation. While the effect on the stroma remains rather speculative, it is a less critical point within the broader scope of this study. It remains open if HGF effect is through epithelial IFN γ receptor expression or through independent pathways. The newly added data addressing this question hints at the latter, but the authors should state that this issue remains open.

We appreciate the overall positive feedback from the reviewer. We agree with the reviewer's comment concerning the mode of action of HGF on IFN- γ signaling and added the sentence "The mechanism by which HGF interferes with IFN- γ remains unresolved" to the discussion (line 433).

Reviewer3's technical concerns have been adequately addressed.

Point-by-Point Response

Reviewer #2 (Remarks to the Author):

The authors have improved their explanations of the functional differences between organoids and assembloids, which should help readers critically evaluate the data. Their explanation for the discrepancy in KRT20+ cell numbers—that de novo KRT20+ cells replace homeostatic cells due to baseline differences between the models—is plausible. However, it remains a weak point that these models reflect in vivo conditions differently for this key finding. A clearer statement acknowledging the limitations of both models would help mitigate criticism. Experimental evidence comparing these systems in this aspect in a targeted way would strengthen the manuscript.

Given the central importance of the loss of KRT20+/BMP2+ colonocytes in acute colitis, apoptosis staining such as TUNEL should be shown as co-staining with key differentiation markers (e.g., KRT20, BMP2), as suggested by Reviewer 1.

The authors have appropriately softened their claims (e.g., regarding BMP signaling and KRT20+ cell extrusion) and revised their conclusions to be more hypothesis-driven rather than definitive, addressing concerns about overstated interpretations.

The justification for using an unrepresentative image (Fig. 3f) in whole-mount imaging remains insufficient. This may leave doubts about data presentation. To prevent inconsistency and improve transparency, zoomed-out views or additional representative images should be provided in the extended data.

We thank the reviewer for all comments. We were very grateful for the positive evaluation and have now also addressed the remaining concerns:

The reviewer further suggested to perform a double IF staining for apoptotic cells together with a marker for differentiated colonocytes. We have now performed this staining (TUNEL and KRT20) and included the new data into the manuscript (see Extended Data Fig. 1h). These new data now further substantiate our findings.

Concerning Fig. 2g (former Fig. 3f): The reviewer expressed their concern about an inconsistency of the data presented in Fig. 2g (former Fig. 3f) in comparison to other images:

We would like to point out that the images from Fig. 2g and also h cannot be compared with the images of Fig. 2k, because they derive from completely different technical processing. g/h are whole mount images while k represents histological sections from paraffin-embedded specimen.

The whole mount processing is performed in nearly-naïve assembloids, which is helpful to maintain the structure of live cells and maintain their fluorescence. However, the processing requires several washing steps that might wash off the dead cells. With paraffin embedding, we would instead expect some slide compression/shrinkage of the specimens, which makes it easier to visualize the dead cells in the lumen. To follow the reviewer's suggestion, we added further assembloid images of the KRT20 traced cells to Extended Data Fig. 3b.